

# Environment Gradient related Dissimilatory Nitrate Reduction to Ammonium in Huangmao Sea Estuary: Rates and Community Diversity

Ran Jiang[1,2], Shu-Xin Zhang[3], Guo Wei[1,2], Rui He[1,2], Li-Ling Yang[1,2], Xin-Sheng Chai[4]

[1]The Pearl River Hydraulic Research Institute, Pearl River Water Resources Commission of the Ministry of Water Resources, Guangzhou, 510640, China

[2]Key Laboratory of the Pearl River Estuarine Dynamics & Associated Process,Ministry of Water Resources, Guangzhou , 510640, China

[3]School of Chemical Technology, Guangdong Industry Polytechnic, Guangzhou, Guangdong Province, 510300, China

[4]State Key Laboratory of Pulp and Paper Engineering, South China University of Technology, Guangzhou, 510640, China

*Correspondence to*: Ran Jiang(458750084@qq.com)

**Abstract.** The potential rates of dissimilatory nitrate reduction to ammonium (DNRA) in the sediments collected from Huangmao Sea Estuary (HSE), one of Pearl River Estuaries in China, were investigated. The research covers a one-year period at 20 sites of HSE based on the $^{15}$N isotope and molecular biology analysis. The results showed that the environment gradients of nitrogen pollutants decreased from the estuary to the sea, and the characteristics of terrestrial pollutant export were obvious. $^{15}$NH$_4^+$ was detected in all sites in continuous-flow systems, which showed that DNRA existed in HSE potentially. In which the maximum $^{15}$NH$_4^+$ was 1948.5ug/L, accordingly in situ rate of DNRA was 6.3 ug/L.h in 72h. The high transcripts of *nrfA* gene were found at the same site, with values of 1159715 copics/(g wet sediment). Correspondingly, a very low transcripts of *nrfA* were found at other sites. Principal components analysis (PCA) based on community composition at genus level showed the distribution pattern of the *nrf*A gene sequences. Nine samples formed three distinct clusters, corresponding to their geographical locations, which suggested that salinity was likely to affect the selection of the *nrfA* bacterial populations. This study provide a better understanding of DNRA in the Pearl River Estuary.

Key word: DNRA; Sediment; Rate; Distribution pattern

## 1 Introduction

Huangmao Sea Estuary (HSE) is one of the estuaries located the western part of Pearl River Delta complex (as shown in Fig. 1), a fastest growing economic area in the southern China. HSE is composed of a bay proper, a tidal river, and several island chains (Wei and Wu, 2011). As the marine quality bulletin of Jiangmen city where HSE locate in reported (Jiangmen Marine and Fisheries Bureau, 2016) inorganic nitrogen (including $NO_3^-$, $NH_4^+$ and $NO_2^-$) was one of the major





eutrophication pollutants in HSE. Since the nitrogen related eutrophication has become a severe problem in the Pearl River

Delta complex area, it is important to have an understanding of nitrogen cycles in this area. From which the good strategy for

the eutrophication control and ecological restoration can be made.

There are several ways to remove inorganic nitrogen from the ecosystem in estuary (Herbert, 1999). Denitrification is regarded

as dominant process for the nitrate reduction in the shallow marine sediments, in which nitrate was eventually forming nitrogen

and released from water body (Herbert, 1999). However, an alternative pathway, i.e., dissimilatory reduction of nitrate to

ammonium (DNRA), could also take place under the certain conditions (Herbert, 1999). As reported, DNRA may be an

important process in the sediments with high inputs of labile organic substrates and limited nitrate availability (Megonigal et

al., 2004). It is often thought to occur when $NO_3^-$ inputs exceed the availability of carbon substrate for denitrification (Tjedje,

1998). Unlike denitrification and anammox, the transformation from nitrate to ammonium via DNRA could prolong the

residence time of fixed nitrogen and thus has a negative impact for removing the bioavailable nitrogen from the ecosystem

(Megonigal et al., 2004).

At present, most of studies on the DNRA pathways were relied on the correlations between the DNRA rates and

environmental factors, such as salinity, temperature, concentrations of $NO_3^-$, organic matter. For example, Laverman et al.'s

work showed that the increase of salinity may, at least temporarily, increase the efflux of $NH_4^+$ from the sediment to the

surface water, which limits the ability of denitrification to remove bioavailable nitrogen from aquatic ecosystems (Laverman

et al., 2007). In Gardner and McCarthy's work, they found that the high salinities, temperature, and organic carbon

availability may exacerbate the effects of nutrient inputs by maintaining bioavailable N in the system as the form of $NH_4^+$

or $NO_2^-$ via DNRA or DNRN (dissimilatory $NO_3^-$ reduction to $NO_2^-$ ), rather than having it to be removed via

denitrification or anammox (Gardner and McCarthy, 2009). Hardison at el.'s work indicated that high organic C loading is a

prerequisite for DNRA to be favored over denitrification, but the $N_2$ production may still be significant when organic C is

high depending on $NO_3^-$ availability (Hardison at el. 2015).

There are some studies available on the organisms capable of DNRA in the coastal sedimentary environments (Gardner et

al., 2009, Gardner et al.,2006) although it is difficult to determine the exact organisms involved. The functional gene assay

(Smith et al., 2003, Takeuchi, 2006), i.e., using a nitrate ammonification related gene -- *nrfA* (named after nitrite reduction

by ferment, coding for another dissimilatory nitrite reductase containing cytochrome c552 [EC 1.7.2.2]), first developed

from anaerobic wastewater treatment reactors in 2004 (Mohan, et al. 2004), maybe an effective approach to be used in such

investigations. By checking the distribution and diversity of *nrfA* gene encoding DNRA in the sediments, it was found that

most of the *nrfA* sequences from the estuarine sediments of Clone River (UK) have several distinct clusters including known

nitrate ammonifiers, e.g., *Aeromonas*, *Shewanella, Desulfovibrio* and *Sulfurospillum* (Takeuchi, 2006). This is also the first



trial to describe the *nrfA* partial sequences derived from a natural environment, with reference to their habitat-specific community structure (Mohan, et al. 2004).

Some investigations have been also conducted for the sediment from different coastal areas in China. For example, the study showed that the N-loss as $N_2$ was the main pathway in East China Sea (ECS), in where DNRA counts for 20–31% of benthic

nitrate reduction (Song et al., 2013). The investigation on the sediments from Yellow River Estuary (YRE) indicated that variability in the activities and community structure of DNRA bacteria (Cuina et al., 2017). There were several reports for showing the coexistence of denitrification and DNRA in Yellow River and Changjiang River in ECS. However, there are the significant differences in both geographic location and climatic conditions between the Pearl River Estuaries (PRE) and ECS or YRE or CRE. Unfortunately, up to date there is lack of studies about the microbial mediated nitrate reduction pathways,

especially DNRA pathway, for the sediments from PRE. Therefore, it is necessary to perform a genetic investigation based on the sediments collected from PRE areas, in order to have a better understanding of DNRA pathway in PRE.

In this work, we chose HSE, one of the PRE, as the model area in the present investigation. The major focuses were to examine the potential rates of DNRA in the sediments collected from the HSE continental shelf; develop the *nrf*A specific primer pairs for describing the distribution and phylogenetic status of the nitrate ammonifiers inhabited in the sediments.

From which not only the environmental-gradient related DNRA profile would be revealed but also the activity and community compositions for the DNRA bacteria in these sediments were identified.

## 2  Materials and methods

### 2.1 Details of the area selected

HSE (see Fig. 1) referred to as Huangmao Bay with an area of approximately 540 km². It is composed of a bay proper in the

lower portion and a tidal river in the upstream. The average water depth in the Bay is approximately 4.5 m. The Bay width changes from 35 km at the mouth to 1.9 km at the head, forming a funnel-shaped estuary(Wei and Wu, 2001). The Bay mouth features a chain of rock islands (Gaolan, Hebao and Dajing Islands) that provides a shelter from wave attack, resulting in an ideal place for port construction and tourism development. There exists a deep navigation channel extending from the Bay mouth towards the Bay head, with water depths ranging from 6 m to 22 m. The study sites were located the Bay mouth, around

Dajing Islands. C19 and C20 were near the west boundary of the Pearl River Estuary Chinese White Dolphin National Nature Reserve.

### 2.2 Sample collection and preparation

Sediments were collected at twenty sites (Fig. 2) during a cruise from Jun 8 to 22, 2016. Each sampling series involved taking three sediment cores from every station. Undisturbed sediment cores were collected by means of plexiglass tubes,

driven directly into the sediment. After closing the tubes with a rubber stopper at the bottom, they were placed in a



thermostatic container, which enabled us to preserve natural conditions, and transported from the sampling sites to the laboratory. During transportation, the container was filled with a 20-centimeter layer of overlaying water from the sediment, which was a source of oxygen and nutrients(An, et al. 2002).

After return to lab, each intact core was installed into a continuous-flow system (Fig.1) consisting of an intake water vessel,
flow tubes, intact sediment core, peristaltic pump, incubation bath, and sample collection vessel (Wayne, et al., 1995). A plunger, with Teflon inlet and outlet tubes and sealed to the core cylinder with an o-ring, was installed over each sediment core and adjusted to 5 cm above the sediment surface to leave an overlying water volume of about 230 mL. Each core was incubated at near in situ temperature and under laboratory light conditions, and site bottom water was passed over the core surface continuously at 1.5 mL min$^{-1}$. After one day the systems was allowed for to approach steady-state conditions, inflow
and outflow samples were collected daily for dissolved gas and nutrient analyses. In the same time the flow water was spiked with 50mg $K^{15}NO_3^-$ , then Inflow and outflow waters were sampled for gas and nutrient analyses after 12 hours, 24 hours, 48 hours, 72 hours, 96 hours, 120 hours, 144 hours and so on. These control and nitrate experiments thus evaluated potential DNRA and offered the advantage of shorter total incubation times to examine N dynamics.

### 2.3 Chemical analysis

Concentrations of $^{28}N_2$ , $^{29}N_2$ , $^{30}N_2$ and Ar were measured with GasBench-isotope ratio mass spectrometry (GasBench-IRMS, CTCAnalyticsAG, Zwingen, Switzerland) which had analytical capabilities of both dual-inlet and continuous-flow modes coupled with an Agilent 6890 GC equipped with a CTC Analytics CombiPAL autosampler, to differentiate denitrification of internal sources of N versus that derived from added $^{15}NO_3^-$ in the overflowing water. The concentrations of $^{29}N_2$ and $^{30}N_2$ were calculated following Holtappels et al. (Holtappels, et al., 2011). Total $NH_4^+$ and $^{15}NH_4^+$ (potential DNRA)
production rates were measured by HPLC (Wayne, et al., 1995).

### 2.4 Rate calculations

The denitrification rates of added $^{15}NO_3^-$ (D15) and *in situ* $^{14}NO_3^-$ (D14) were calculated from the measured production of $^{14}N^{15}N$ (p$^{29}N_2$ ) and $^{15}N^{15}N$ (p$^{30}N_2$ ): D15= p$^{29}N_2$ + 2p$^{30}N_2$ ; D14 = (p$^{29}N_2$ /2 p$^{30}N_2$ )·D15 (Gruca-Rokosz, et al.). The total *in situ* rate of dissimilary $^{14}NO_3^-$ reduction to $NH_4^+$ (DNRA) was estimated from the production rate of $^{15}NH_4^+$
(p$^{15}NH_4^+$) in anoxic–anaerobic $NO_3^-$ reduction zone. Assuming that DNRA takes place in the same sediment stratum as denitrification, the $^{15}N$ atom% of $NO_3^-$ reduced to $NH_4^+$ is the same as the $^{15}N$ atom% of $NO_3^-$ reduced to $N_2$. The in situ rate of DNRA can therefore be calculated as: DNRA = p$^{15}NH_4^+$·(D14/D15)(Gruca-Rokosz, et al.,2009, Giblin, et al., 2010). Further details are given by Risgaard–Petersen and Rysgaard (Risgaard-Petersen, and Rysgaard, 1995) Partitioning of DNRA and $N_2$ production was expressed as %DNRA (i.e., %DNRA = 100[DNRA/(DNRA + $N_2$ production)]), with total $N_2$
production in the model or p$^{30}N_2$ and $^{29}N_2$ in the vials accounting for the remainder(Hardison, et al. 2015)





## 2.5 Molecular biology analysis and DNRA communities

**DNA Extraction from the sediments.** Bulk sediment samples were collected in sites C1, C3, C5, C8, C11, C13, C15, C16 and C17 from upper (0–3 cm) layers using an ethanol washed spatula, and stored in crashed ice. DNA was extracted from wet sediment samples (1.0 g) using the PowerSoil DNA Kit (MO BIO Laboratories, USA) according to manufacturer's

instructions. DNA concentrations were measured on a spectrophotometer (E-Gel Imager, U.S.).

**Design of PCR primers**  Primer pairs (6F/5R and 6F/4R) were designed based on the existing partial *nrf*A sequences and complete sequences including the nrfA fragment in an open reading frame (ORF). A new pairs of forward and reverse primers(6F/5R) were manually designed based on guidelines (Alkami Biosystems 1999) and the ClustalW (Thompson et al. 1994) multiple DNA sequence alignment. Two conserved regions of the nrfA sequences, consisting of one based on six

sequences (AE008900, AL627282, AE005640, X72298, U32787, AE006035) and the other based on four sequences (AE005640, X72298, AE008900, AL627282) were respectively chosen as the target of designing the new primer pairs, 6F/5R, respectively. Another primer set (6F/4R) developed by Takeuchi (2016) was also employed after the PCR condition was optimized in this study.

Sequences of the nrfA primers used in this study are listed in Table 1. Compared with Primer pairs (6F/4R), the PCR

produces of primer pairs (6F/5R) had more conservative sequences for high-throughput sequencing of nrfA gene in HSE Estuary samples.

**High-throughput sequence of *nrf*A gene .** The DNRA community diversity of DNRA bacteria was determined on an Illumina MiSeq sequencer. Sites C1, C3, C5, C8, C11, C13, C15, C16 and C17 were selected to examine the community structure. After DNA extraction, PCR was conducted in triplicate using PCR Amplifier 2720 under the same conditions as those of Song

et al (Song, et al. 2014). The PCR amplification was performed using a programmable thermocycler, Gene Amp 9700 (Applied Biosystems). The programmers consisted of the following conditions: the initial denaturation at 94∘C for 2 min, 30 cycles of 58°C for the first-round PCR (20cycles for the second-round PCR) for 1 min, followed with an annealing temperature at 46.5∘C for the first-round PCR (at 54.5∘C for the second-round PCR).

**Sequence analysis and phylogenetic classification.** Raw reads of *nrf*A gene sequences were saved in the FsatQ file. First,

the sequences shorter than 150 bp and lower than 20 quality score, as well as the sequences containing ambiguous base N, were removed through the sliding window method. Paired reads were merged based on the overlap of bases using FLASH (http://ccb.jhu.edu/software/FLASH/)( Magoc., 2011). Then, valid sequences were obtained through the identification between the merged reads and the relevant samples. Chimera sequences were removed using QIIME (http://qiime.org/) (Edgar, 2010) to get high quality sequences. Then, they were clustered into OTUs using UCLUST56 with 97% similarity

and rare OTUs containing lower than 0.001% of total *nrf*A sequences were omitted(Edgar, 2011). The sequences with highest abundance of each OTU as representative sequences were annotated with NCBI taxonomy using QIIME (soft version 1.9.1). Microbial community richness indices included Chao1 estimations, while diversity was quantified using the Simpson and Shannon indexes (Edgar, 2011). The alpha diversity index of each sample was determined using QIIME after a





flattening process sequence(Page, 1996). Rank abundance curve was drawn using R software. The length of X-axis represented richness and slope of lines represented evenness. A steep gradient indicated low evenness. The differences of community composition among four sites along the salinity gradient were evaluated through PCA based on community composition at genus level using QIIME software. Distributions of DNRA bacteria in four samples at different levels were

determined using QIIME, and a heat map was constructed with the 50 most abundant genera using R software. In addition, RDA was conducted to evaluate the relationship between environmental factors and community structures using R software. Phylogenetic trees were generated using the neighbour-joining method as implemented with a TreeView programme (Page, 1996).

## 3 Results and discussion

**3.1 Environmental Gradients of the Estuary**

Bottom seawater and sediment characteristics were investigated at 20 stations (Fig.2). We have carried out one year in C1-C20 sites for sampling quarterly. The sea water quality standard, i.e. GB3097-1997, were used in assessing the sea water quality of HSE. Dissolved inorganic nitrogen (DIN) referred to the totality of $NO_3^-$, $NO_2^-$ and $NH_4^+$. According to one year research, the data showed that $NO_3^-$, DIN and $NH_4^+$ in C1-17 sites exceeded the standard four level with 80% datas. Most of

sites except C18-20 sites were evaluated in the sea water quality standard four level, while C18-20 sites were evaluated in three level where located near the Pearl River Estuary Chinese White Dolphin National Nature Reserve. As is showed in Fig. 3a, DIN in the bottom water decreased sharply from the estuary coast ($\sim$ 0.842 mg/L) to the outer shelf($\sim$0.270 mg/L). In the same time, $NO_3^-$ range in 0.17-0.567mg/L. $NO_3^-$ was accounted for 60-80% of DIN. About 80% sites accessed 0.5mg/L DIN. In most of time, C18-20 have good sea water quality. The maximum concentration of DIN appeared at the mouth of

Yamen Gate, then decrease from upstream and estuarine coast to downstream. The concentration of DIN is basically higher in the west and lower in the east, as shown in concentration profile figure. The highest concentration of the upper estuary area is above 0.95mg/L, while the NID near the western estuary was above 0.8mg/L, but decline from estuary to outer shelf. One year datas showed that the concentration of nitrogen pollutants decreased from the estuary to the sea, and the characteristics of terrestrial pollutant export were obvious.

From Fig. 3b, the major form of DIN is $NO_3^-$, the pollutants in the western estuary was more serious than that of eastern area because of factories and hydrodynamic force. From figure 3c, the concentration of $NH_4^+$ showed a rising trend, except C19-20 is rather low. Using China sea water quality standard to evaluate, most of sites access 0.02mg/L which is the maximum concentration in China sea water quality standard four level. According to one year monitoring, $NH_4^+$ in Spring are low while Summer and Winter are high.

As was shown in Figure 4, the salinity was higher than that of the west, and the salinity line inclined to the west, especially in flood season. The analysis of salinity characteristics was consistent of the hydrodynamic characteristics of the eastern tidal





current in HSE. In flood season, the salinity was affected by fresh water runoff, so it was less than in the dry period. The salinity of our research area was near the salinity Contour(17000 cl⁻ mg/L) in bottom water in the North of Dajin Island.

## 3.2 The potential rates of DNRA

$^{15}NH_4^+$ was detected in all sites after 12h in continuous-flow systems, which showed that DNRA existed in continuous-flow

system. $^{15}NH_4^+$ ranged in 310.0 ug/L-1948.5 ug/L after 216 hours experiments, but in situ potential rate of DNRA was different in every sampling sites. Moreover the maximum $^{15}NH_4^+$ was 1948.5ug/L in C3, accordingly in situ rate of DNRA was 6.27ug/(L.h) in the end. The second maximum $^{15}NH_4^+$ was 1703.1 ug/L in C17, accordingly in situ rate of DNRA was 2.75 ug/(L.h). While the low $^{15}NH_4^+$ was 310.0 ug/L, 398.2 ug/L in C8 and C16, accordingly in situ rate of DNRA was 1.53 and 2.36 ug/(L.h). The in situ rate of DNRA increase with the growth of cultivation time, and basically smooth in five days, which

reach the peak of about 23.3 ug/(L.h) in one to three days. Then the rate slowed down after five day in the range of 0.28-13.4 ug/(L.h). Sample sites have different hydrodynamic and pollutant characteristics, C1 and C11 were in the sites where the upstream pollutant were more likely to pass, exposing to incoming sea pollutants, while C5 and C7 were relatively near the coastline. As were shown in Fig.5a and 5b, the potential DNRA rate didn't prove any significant statistical differences between particular stations in salinity range. it is believed that the most important factors effected DNRA were related to the living

environment of related functional microorganisms. Furthermore temperature, salinity, nitrogen and phosphorus may also be important factors regulating the fate of $NO_3^-$. Due to the research on the role of DNRA in nitrogen conversion in PRE, this reasearch can be used to supplement the active nitrogen pollution pathway in the southern China.

## 3.3 DNRA rates and nrfA gene abundance

The abundances of DNRA bacteria based on *nrfA* gene quantification in the sediments of 9 sampling sites in Supplementary

Table S1. The two *nrfA* primer (6F/4R) and (6F/5R) were used in high-throughput sequencing respectively. Effective tags of 6F/5R *nrfA* primer ranges about in $11\times10^6$ to $7\times10^5$, while which of 6F/4R *nrfA* primer ranges in $7\times10^5$ to 9 . So the analysis of DNRA community based on 6F/5R *nrfA* primer. DNRA rates significantly among 0.13 ug/L.h to 23.3 ug/L.h in 72 hours. The highest DNRA rate was found in C3, which is locate near coastal pollution zone, while lower rates were measured in the sediment samples of sites C16 and C18 which have low DNRA rates less than 3 ug/L.h in 72 hours . High transcripts of *nrfA*

gene were found at C3 , with values of 1159715 copics/(g wet sediment), respectively. Correspondingly very low transcripts of *nrfA* were found at C8, C16 and C15.

## 3.4 Spatial Distribution Patterns in the Sediment

Via high-throughput sequencing, raw sequences were obtained from the sediment samples of Pearl River Estuary. After the removal of low quality reads, a total of 80,407–1,196,456 qualified sequences were produced. Then, a total of 74,173–



1,159,715 effective sequences for these samples were obtained via further filtering, which were clustered into operational taxonomic units (OTUs) ranging from 2,541 to 8,709 (Supplementary Table S2). As shown in this table, site C3 shared the highest number of OTUs than any other sites, while site C8 and C16 shared few OTUs with other sites. Alpha diversity indices including Rarefaction Curves, Chao1 Curves, Shannon Curves, Simpson Curves and rank abundance Curves are

listed in Supplementary Fig. S1. The Chao 1 estimators showed that the highest total species richness occurred at C3, while the lowest richness occurred in the farthest from the coast at C16. Both the Simpson index and the Shannon index of the sediment samples of sites C15 and C5, C17, C13 were higher than those of the other sites, which indicated that higher community diversity was found in these four sites, though the difference was not substantial. Similarly, the same trend of sequence evenness with community richness was found (Fig. S1-e). The line representing site C3 was the smoothest with the

highest total species richness, and site C15 had the highest community evenness.

To compare the DNRA communities of nine sites, principal components analysis (PCA) based on community composition at genus level was performed (Hamady, 2010). Figure 6a shows the spatial variation of nine samples with 54.36% (PC1) and18.15% (PC2) of the variance explained. From Fig 6a and 6b, It indicated that site C5 and site C15 were clustered together, and the community structures of these two sites were similar. The community composition of site C11 also shared

some similarities with those of sites C17.  C3,C1,C8 and C16 were divided in one group in some similarities. However, the community of site C13 was the most distinct of all the samples.

The distribution pattern of the *nrf*A gene sequences from the surface sediments (0–1 cm) in HSE estuary was investigated. As shown in Figure 6b, the nrfA sequences distributed in the nine sites covering a 10000-20000 cl$^-$ mg/l range of salinity formed three distinct clusters corresponding to their geographical locations. From figure6 PCA, nine samples were divided in

three group which were in the same sampling place. The sequences collected from C1, C11 and C13 were clearly separated from the other cluster. C8 and C16 were just in the same salinity contours which were still tilted west, This result suggests that salinity per se was likely to affect the selection of the *nrfA* bacterial populations.

In contrast with the past studies aimed at the phylogenetic analysis in pearl river estuary, this study was focused on the specific community responsible for nitrate ammonification. Since the *nrfA* sequences had been so far extracted from various

sources (bioreactors and estuarine sediments) as well as from laboratory cultures beyond the phylogenetic groups (*α-,γ -, δ-, ε-proteobacteria* and the *Bacteroides-related* subdivisions), it was quite reasonable to detect the similar nrfA sequences of different phylogenetic origins in the environmental samples (e.g.,one from the *Enterobacteriaceae* and the other from sulphate reducing bacteria). This approach makes it possible to detect the functional marker in situ but may reduce its analytical resolution into the community structure. As a result, the *nrfA*-based grouping was not able to distinguish the

freshwater from seawater sequences but successful in extracting the estuarine sequences from the others.

## 3.5 Diversity of the nrfA Sequences in the Sediments

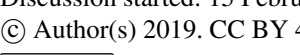



After removing rare OTUs, modified OTUs were classified into different levels such as phylum, class, order and so on. The classification results are listed in Supplementary Table S3. In this study, a total of 7 bacterial phyla were identified. Dominant phyla (>0.5% sequence abundance in at least one site) are listed in Fig. 7a. For bacterial phyla, *Proteobacteria* was the most abundant phylum in all samples, accounting for 99.93–91.56%. In addition, many sequences based on *nrfA*

pyrosequencing were not classified to certain type, range from 8.1% in site C5 to 0.07% in site C16,and they appeared in sediment samples from home and abroad.

At the class level, as shown in Fig. 7b, a total of 8 classes in eight sites were obtained. *Gammaproteo* bacteria was the most abundant of all sites, accounting for 99.9–90.5%. other *Proteobacteria* such as *Alphaproteo* bacteria and *Deltaproteo* bacteria are existed but less than 0.5%. As depicted in Fig.7c, *Enterobacteriales* (89.9-99.9%) which belong to

*Gammaproteo* bacteria was the most abundant order. Of all filtered sequences, a total of 11families were detected. *Enterobacteriaceae*, which is member of *Enterobacteriales*, were more abundant than other families, accounting for 89.9-99.0% of the classified sequences. *Microbulbiferaceae* belonging to *Cellvibrionales*, *Moraxellaceae* belonging to *Pseudomonadales*, and other 7 families were present in site C5 at low levels. While other site have only 1-3 families.

An overview of the diversity of the *nrfA* sequences in the HSE sediments was also examined. A total of 63 sequences

retrieved from the sequences found in the sediments were compare with those from known strains and environmental isolates. Fig. 8 showed a dendrogram, suggesting the relationship of partial nucleotide sequences form the *nrfA* sequences and the other reference sequences obtained from the databases.

Three nucleotide sequence clusters were roughly grouped: One cluster belonged to the known group as nitrate ammonifiers with the *nrfA* sequences, comprised of fermentative bacteria (*Escherichia coli*)( Cole, et al., 1980) and the others. Most of

the *nrfA* sequences from the HSE sediments were closely related to the *nrfA* sequences from *Shewanella*, and *Sulfurospirillum,* which are known members of the nitrate ammonifiers(Takeuchi, 2006). A few of the sequences were quite divergent from the others and affiliated with the *nrfA* sequences retrieved from versatile anaerobes such as sulphur-reducing nitrate ammonifier *Sulfurospirillum deleyianum*(Eisenmann, et al. 1995). These microbe seem to be favored anaerobic marine sediments full of iron sulphide, involved in sulphur and iron cycles coupled with nitrate ammonification. Although

the present result partly supported the previous view on the contribution of fermentative bacteria to nitrate ammonification, it also suggested the majority of the actual agents could be other metabolically versatile but less described groups such as *Shewanella*, *Sulfurospirillum* and *Bacteroides*(Cuina, et al., 2017, Takeuchi, 2006). Widespread distribution of these *nrfA* sequences found in this study could b e a result of their adaptation to nitrate and organics rich habitats. The nitrate ammonifying populations in the sediment have been so far treated as a minor component but ecologically important not only

as a consumer of nitrite but also as a producer of ammonium. while some formed unique clusters grouped with *Dusulfovibrio salexigens* or supposedly uncultured members of bacteria.

**4 Conclusion**




One year research in the 20 sites of HSE shown the environment gradients of nitrogen pollutants decreased from the estuary to the sea, and the characteristics of terrestrial pollutant export were obvious. This study investigated the existence of DNRA using the 15N isotope paring method in continuous-flow systems. The maximum $^{15}NH_4^+$ was 1948.5ug/L, accordingly in situ rate of DNRA was 6.3ug/L.h in 72h. High transcripts of *nrfA* gene were found at the same site, with values of 1159715

copics/(g wet sediment), respectively. Correspondingly very low transcripts of *nrfA* were found at other sites. The distribution pattern of the *nrf*A gene sequences from the surface sediments formed three distinct clusters corresponding to their geographical locations. which suggests that salinity per se was likely to affect the selection of the *nrfA* bacterial populations. Base on nrfA gene pyrosequencing, *Enterobacteriaceae*, which is member of *Enterobacteriales*, were more abundant than other families, accounting for 89.9-99.0% of the classified sequences. This study improves our understanding

of DNRA in the Pearl River Estuary. However ,the contributions of anammox, denitrification and DNRA to nitrate reduction based on activity analysis require further study.

**Acknowledgments**

This study is funded by the National Natural Science Foundation of China (Grant no. 51409287) and the Guangdong Province Natural Science Foundation of China (Grant no. 2017A030313329)

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

**Table 1. Sequences of the nrfA primers used in this study**

| Primer | Sequence(5′ to 3′) |
|--------|--------------------|
| 5R | CGCCAYTGVGCRTGRCGRATATC |
| 6F | GAYTGCCAYATGCCRAAAGT |



| 4R | GCATCCGCSGCTTTATCCAT |
|---|---|

F: forward primer, R: reverse primer;

Base Codes: K=T/G, R=A/G, Y=C/T, S=C/G

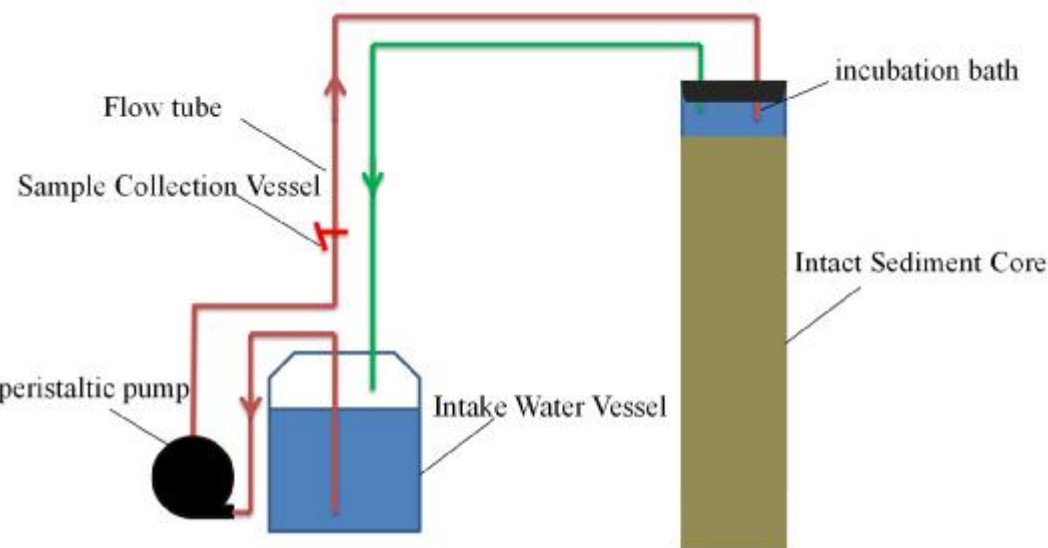

**Figure1. Diagram of Continuous-flow System**

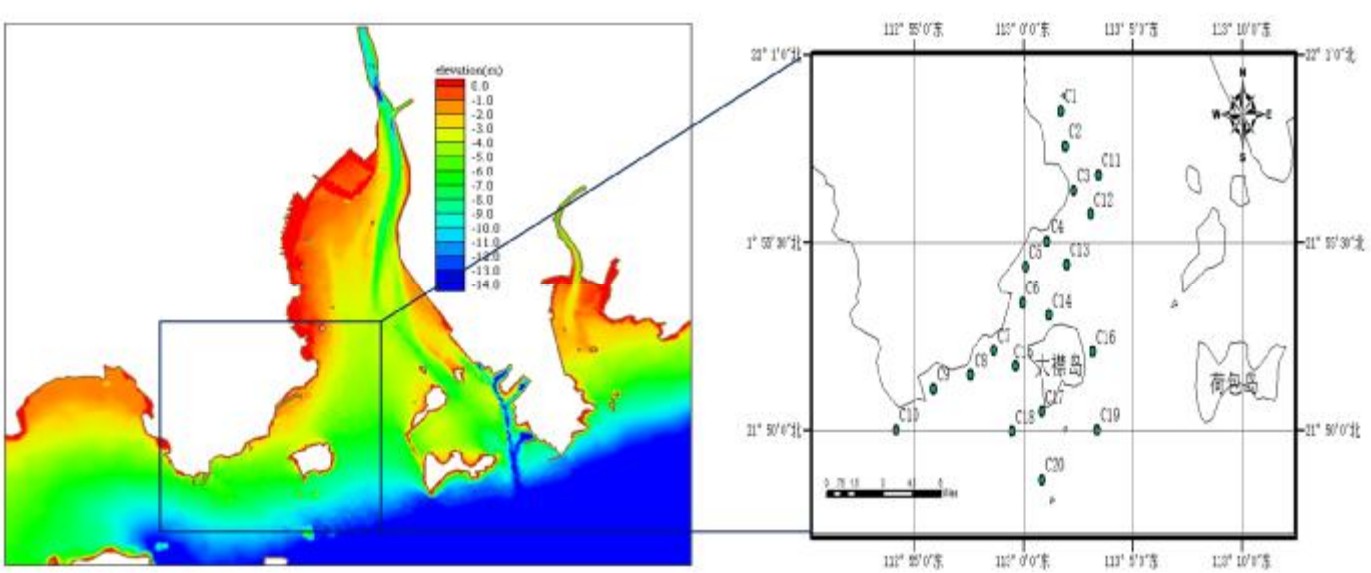





**Figure 2. Location of sampling stations in HSE**

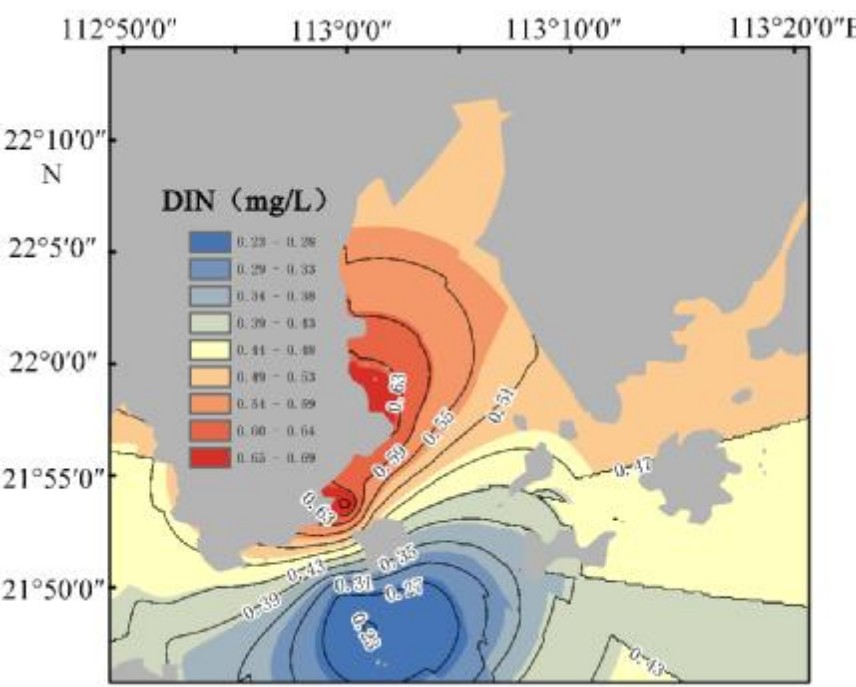

**Figure 3a. DIN Spatial distribution**

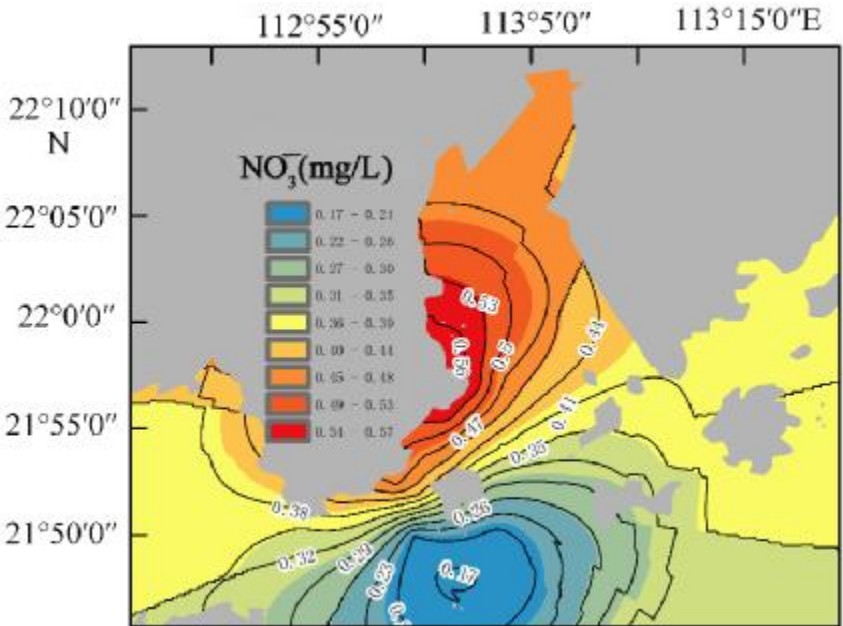

5   **Figure 3b. NO$_3^-$ Spatial distribution**



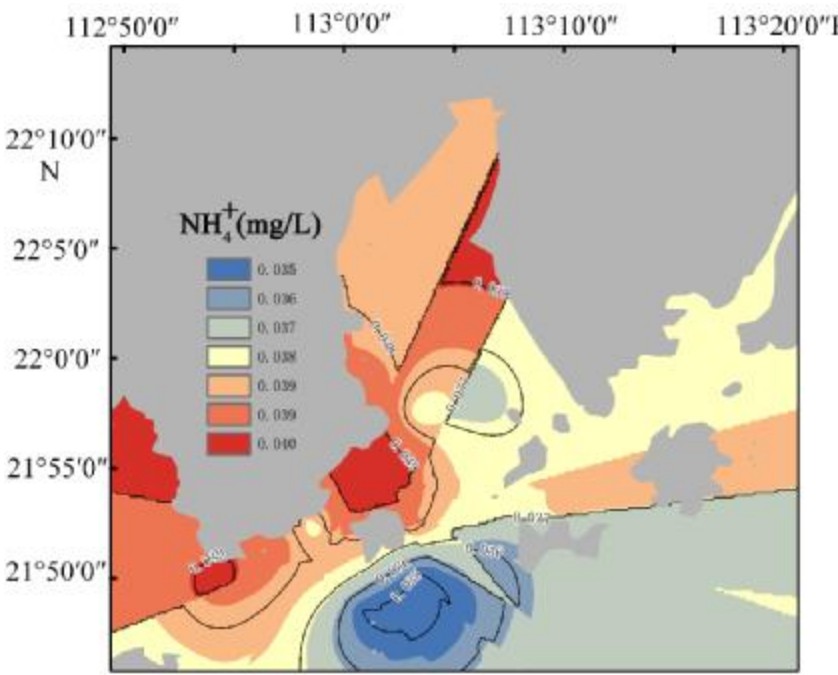

**Figure 3c. NH$_4^+$ Spatial distribution**



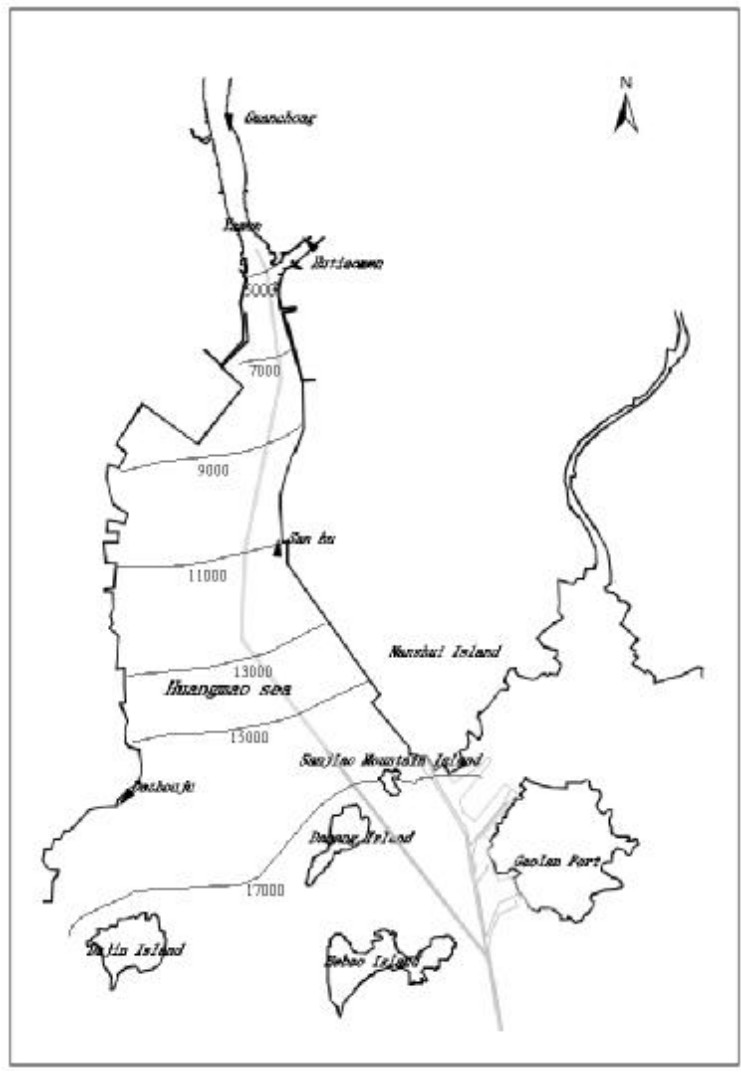

**Figure 4. Flat Distribution of Chlorine Content in the bottom of HSE at maximum tide level, unite cl⁻ mg/L**





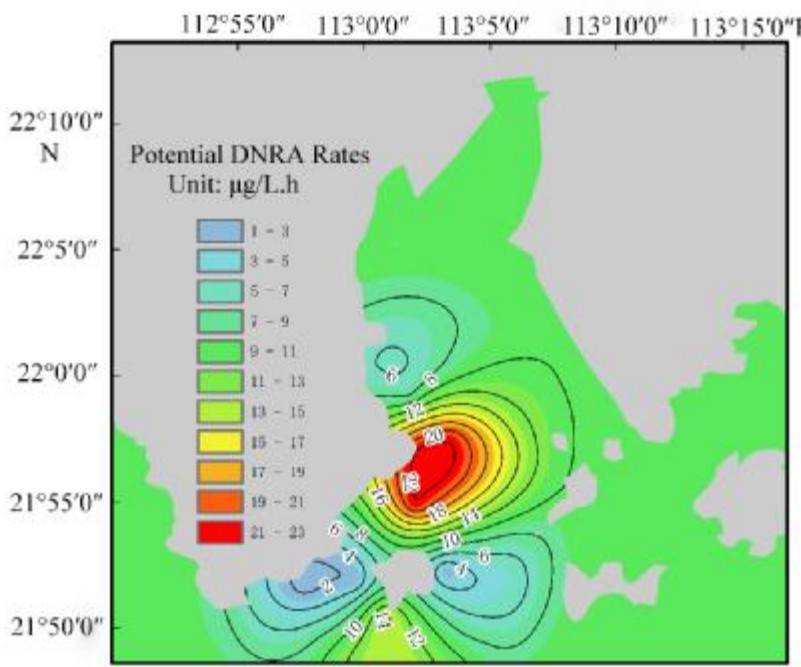

**Figure 5a. Potential DNRA Rates Spatial distribution in 24 hours**

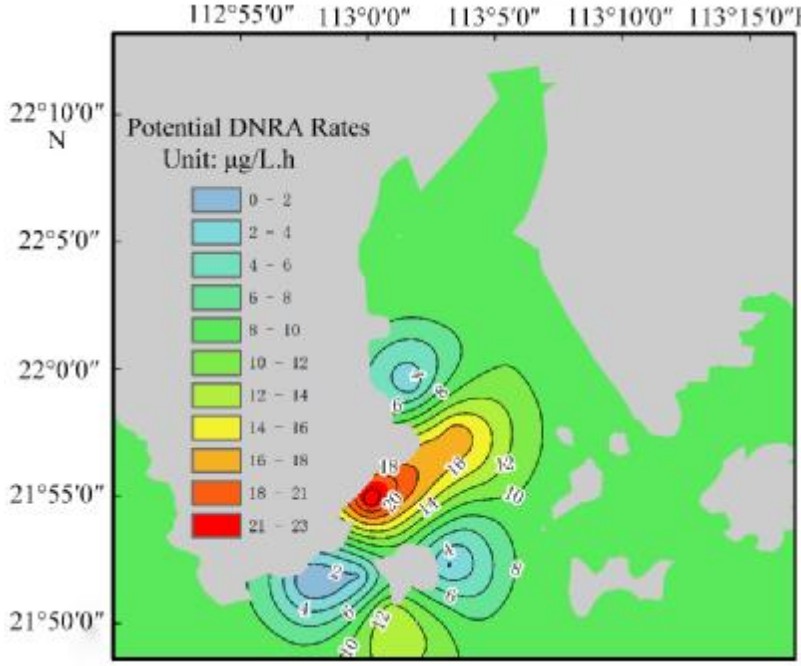

5    **Figure 5b. Potential DNRA Rates Spatial distribution in 48 hours**



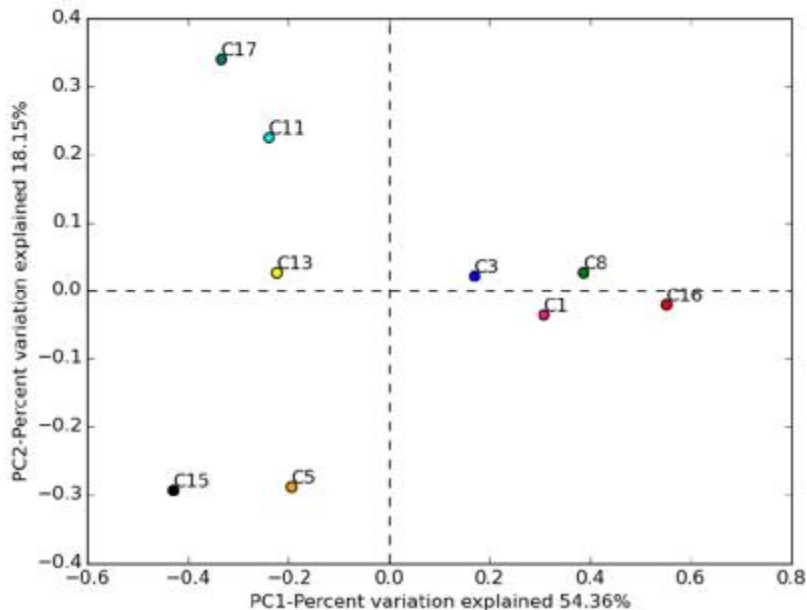

**Figure 6a. PCA analysis of DNRA communities in HSE. Unifrac phylogenetic distance calculation method based on Quantitative Weighted was used to calculate clustering. Percentages represent the amount variance explained by each dimension**

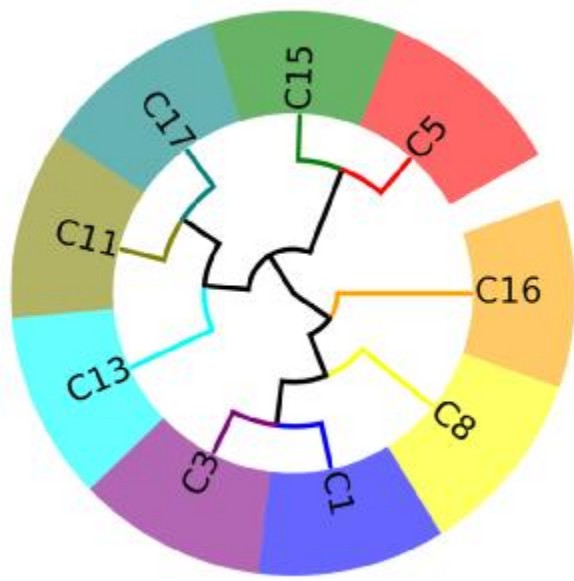

**Figure 6b. Unweighted Pair Group Method with Arithmetic Mean(UPGMA) analysis of DNRA communities in HSE**





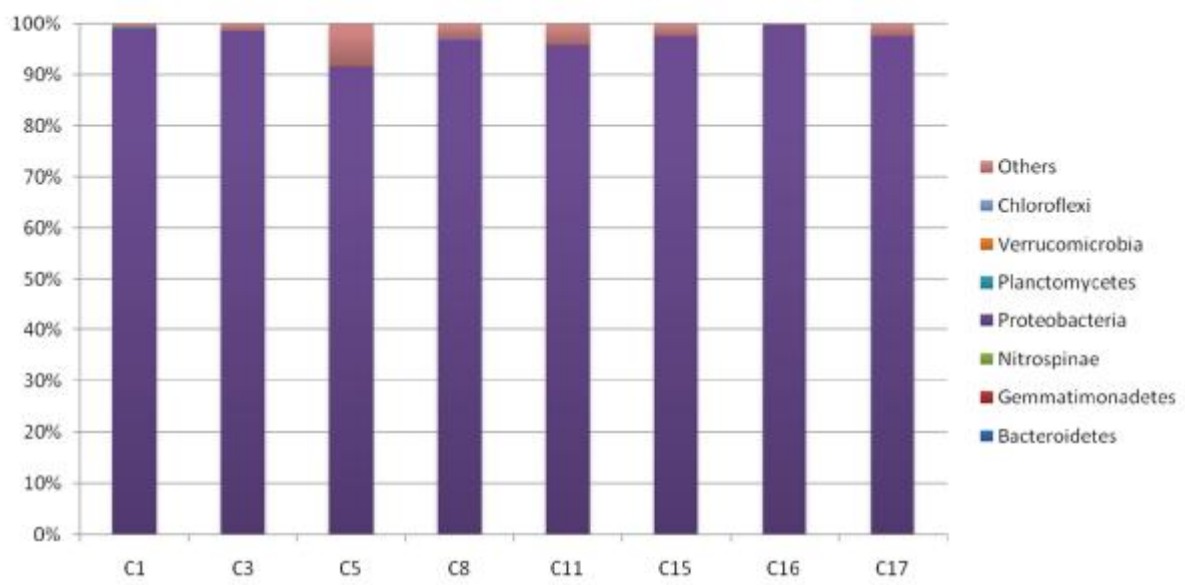

**Figure 7a. Distributions of DNRA bacteria in HSE at the phylum level (b)the class level (c) the order level (d) the family level**

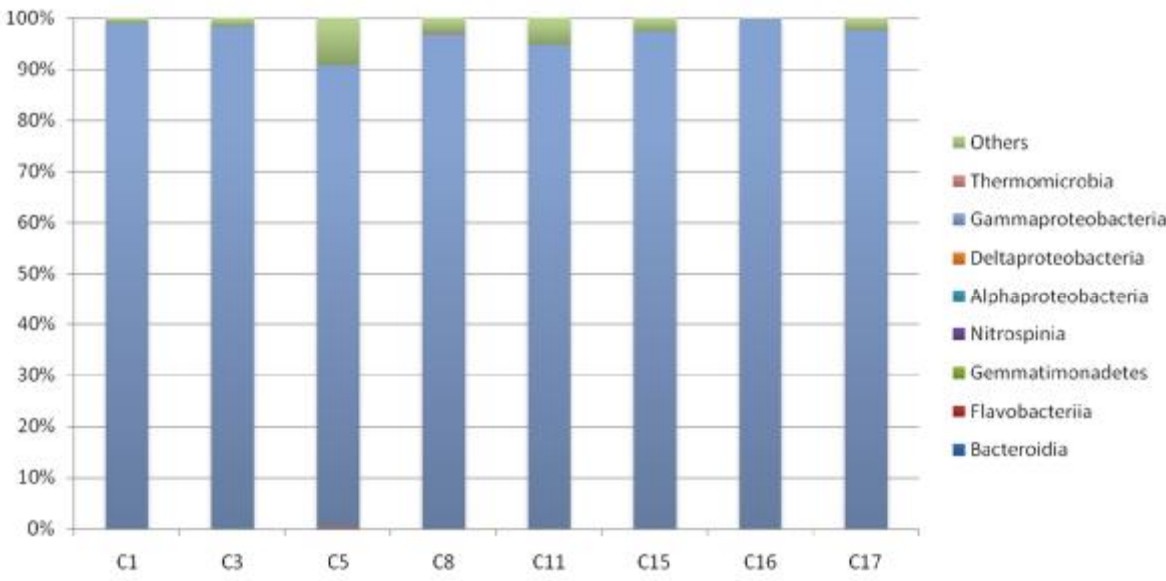

5    **Figure 7b. Distributions of DNRA bacteria in HSE at the class level**





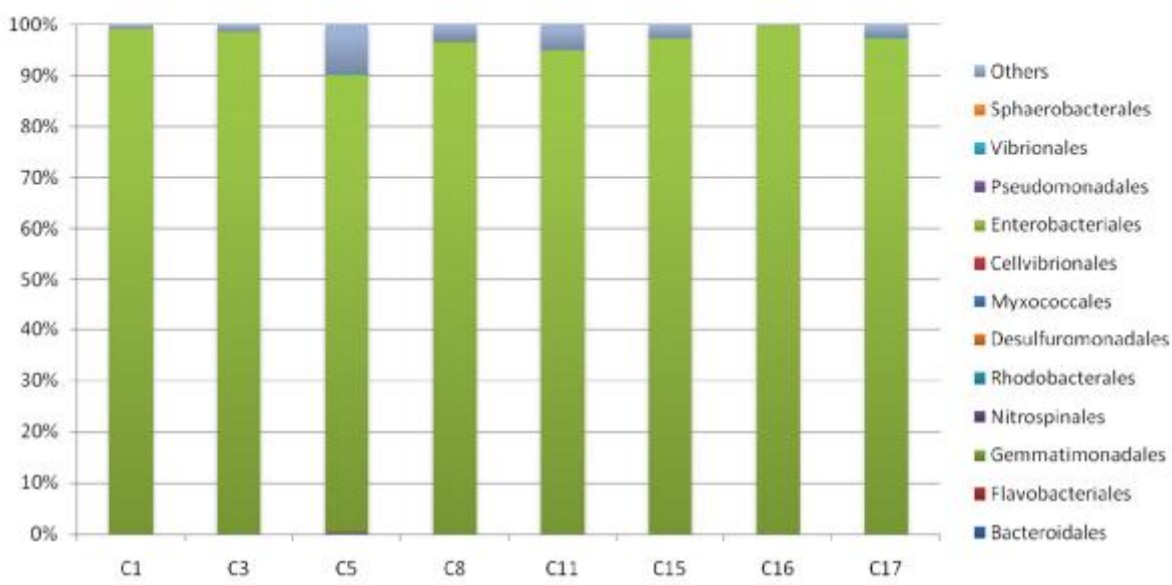

**Figure 7c. Distributions of DNRA bacteria in HSE at the order level**

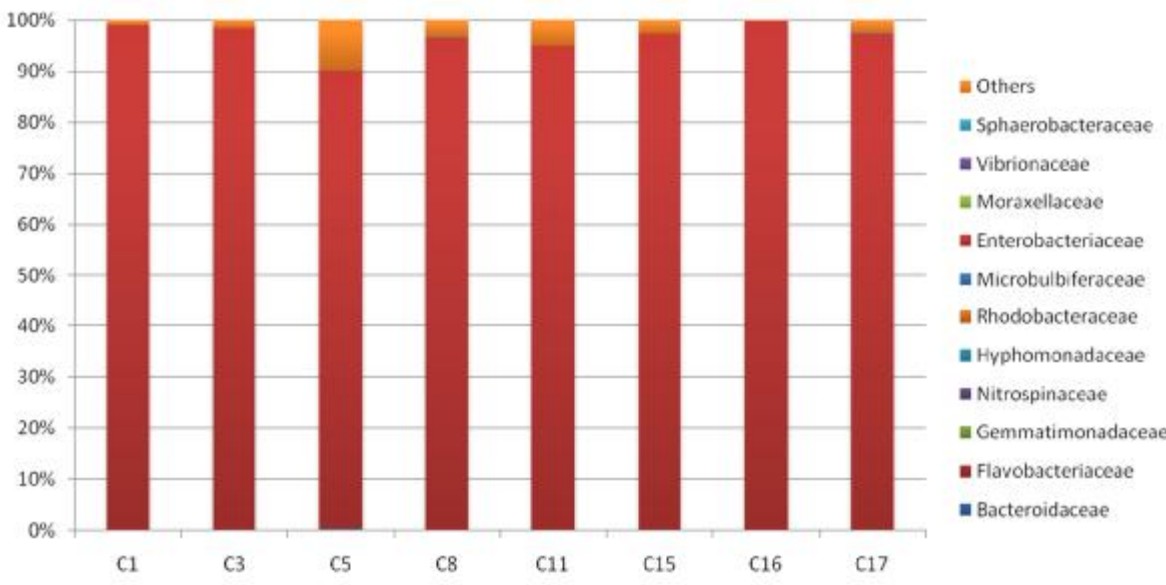

5 **Figure 7d. Distributions of DNRA bacteria in HSE at the family level**




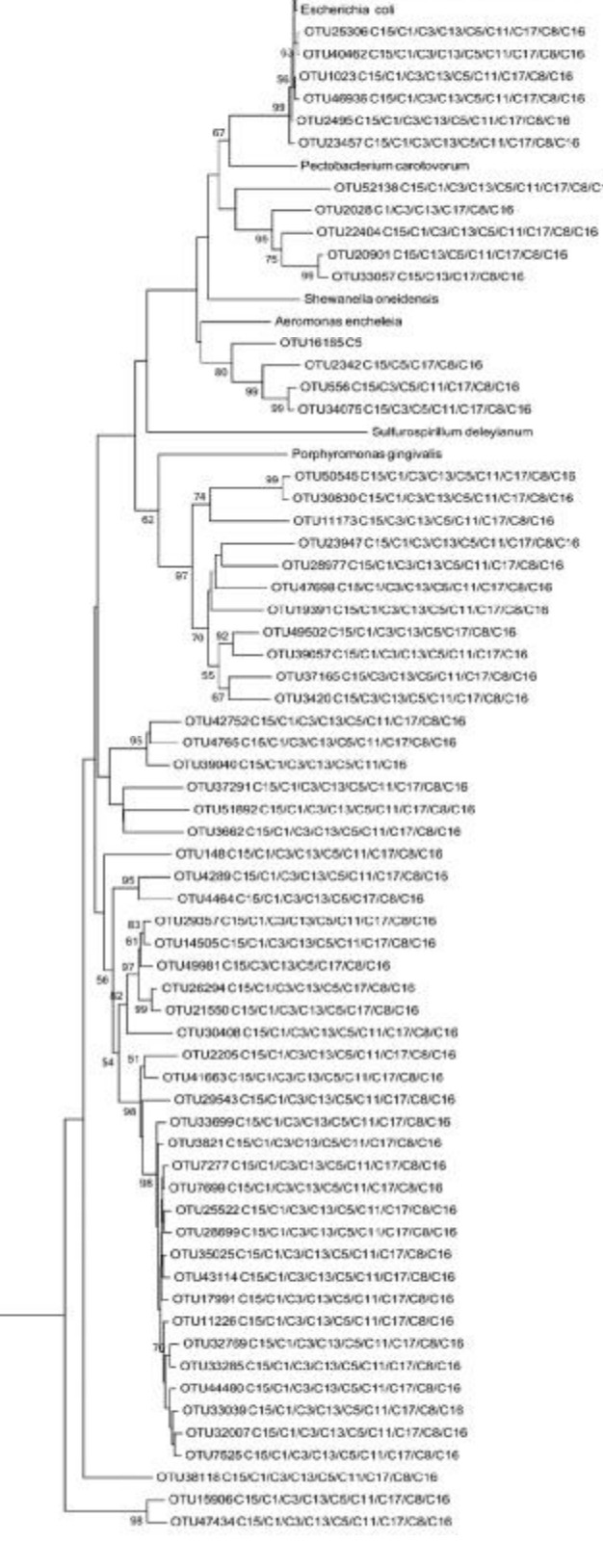





**Figure 8. Dendrogram showing the diversity of the *nrf*A partial nucleotide from the sediments in HSE**, with reference sequences from the databases: *Escherichia coli, Shewanella, Sulfurospirillum, Dusulfovibrio* and so on. The bar indicates 10% nucleotidesequence divergence. Numbers near the branches represent the percentages of 1,000 bootstrap repetitions. Confidence limits of less than 50% are not shown.