# Peer review of "Environment Gradient related Dissimilatory Nitrate Reduction to Ammonium in Huangmao Sea Estuary: Rates and Community Diversity"

_Biogeosciences, 2019_

## Referee Comment (RC1) · Anonymous Referee #1 · 6 Mar 2019

This is a fairly direct study that reports potential rates of dissimilatory nitrate reduction to ammonium (DRNA) in the HSE sediments in China, where the nitrite reductase nrfA gene is determined to infer spatial variation in nitrate ammonifier community structure. A main finding is the potentially widespread occurrence of DNRA and geographical separation of DNRA bacteria community in the study region. While the data in this study might add to our understanding of the complex N cycle in natural environments, there are multiple major issues that render the current form of manuscript unsuitable for the journal of Biogeosciences.

The first issue is a lack of strong research motivations. While the Introduction briefly

reviews the background about DNRA and previous work, it lacks a rational development of research questions, motivations, or working hypotheses. The authors should stress more in the Introduction on what the key unaddressed questions are after reviewing the previous work (e.g., page 2, lines 13-30; page 3, lines 3-7), why they are important, and how the results from the HSE/PRE region would help resolve these unknowns. One way to improve is to start off with a bigger picture of N cycle/unknows (rather than a description of the HSE region), followed by an overview of previous work on DRNA that subsequently leads to major research questions, and then justification on the choice of PRE sites (not just because "there is lack of study . . . from PRE"). Note that the fact that there hasn't been any study in the PRE region is by itself not an attractive incentive because most audiences would probably not be familiar or interested in PRE or HSE region in particular, unless such environment is unique for this kind of research and can provide new insights.

The second major issue is the presentation of data. The manuscript is written in a data-reporting style with very little discussion on the data. The authors are expected to place their results in the context of the existing literature and synthesize their findings into lucid scientific story that advances our understanding of N cycle in a broader sense (i.e., not just for the PRE region). A "real" Discussion section is currently missing from the manuscript. Another issue with data presentation is that most figures are of low quality and it is very difficult to read the number, text, or color (e.g., Figs. 2, 3, 4, 5, 7, 8).

The last issue is the frequent improper uses of English and language that need to be checked thoroughly throughout the manuscript. A few examples are given below.

Other comments Page 1 line 15: Please specify "nitrogen pollutants" and "characteristics of terrestrial pollutant export".

Line 16: The readers would probably not understand what the "continuous-flow systems" is until they read the methods.

Line 17: English. I would suggest replacing "accordingly" with "the corresponding" throughout the manuscript.

Line 25: English. Insert "in" after "located"

Line 27: English. "As the marine..." This sentence has grammatical issues; please rephrase.

Page 2 Line 2: English. "From which the good...". This should be a clause that follows the previous sentence; please check the grammar.

Line 5: English suggestion: change "forming" to "converted to"

Line 13: English. Delete "of" and "were"

Page 3 Line 7: English. Remove "the" from "there are the"

Line 8: The authors could elaborate the geographic and climatic differences between HSE and other study sites (ECS and YRE); that will help justify the choice of the HSE for this study.

Line 19: English. Insert "is" before "refereed"

Line 24: English. Insert "in" after "located"

Page 4 Line 1: English. Replace "transported" with "transport"

Line 12: "and so on". Please list all the sampling time points.

Page 6 Line 12: Please specify the "sea water quality standards, GB3097-1997".

Line 14: What are "the standard four level" and "80% datas"?

Page 7 Line 2: "17000 cl-1 mg/L". Please use the more common unit for salinity (e.g., Practical Salinity Unit) in the manuscript.

Lines 3-26: The presentation is quite descriptive. Same issue is found on page 8. It's partially because the presentation of data is frequently referred to specific sites

(e.g., C1, C7, C11), which have limited meanings to the general audience. Just as an example, a description of "higher XX rates were found at Stns 1, 2, 3..." is less interesting/meaningful than a description of "higher XX rates were found in lower-salinity regions (such as Stn1, Stn2, Stn3)". I strongly suggest to separate the Results section from the Discussion section.

Line 22: English. "rates significantly among". Please check the grammar.

Line 23: English. "locate" should be in past tense.

---

## Referee Comment (RC2) · Anonymous Referee #2 · 20 Mar 2019

The authors examined the potential rates of dissimilatory nitrate reduction to ammonium (DNRA) using time-series incubations with supplemental 15N-NO3-, and applied high-throughput Illumina sequencing technology to assess the distribution and diversity of the nitrate ammonifiers nrfA genes in the surface sediments collected from the Huangmao Sea Estuary (HSE) continental shelf. There are many too bad problems in this manuscript. The data and analysis are too rough. The potential rates of DNRA are not reliable. I am not convinced for its publication in BG and even BGD.

1) The English of the paper needs an overhaul. Most of the sentences are wrong, unclear, lacking logic, or misleading. Some references cited in the main text are missing

in the list.

2) The methods were not described clearly. Many key details are missing. For example, the one-year observations were not stated; the biogeochemical parameters (temperature, salinity, nutrient concentrations, etc.) measurements are missing; the procedure of the high-throughput sequencing of the nrfA gene is incomplete...

3) The method on time-series incubations with supplemental 15N-NO3- to examine potential rates of DNRA needs to clarify the cross-feeding effect since heterotrophic bacteria can uptake 15N-NO3- and release 15N-NH4+ to the experiment system.

4) There is no statistical analysis in this manuscript. All statements and conclusions on the control of environmental parameters on communities are not supported by any specific analysis or statistical test.

5) The whole paragraph of "2.4 Rate calculations" (Page 4, Line 22-30) was plagiarized from the literature. Also, some statements in the Introduction section (Page 2, Line 17-22) were from the literature directly.

6) Very few discussion was made in the Results and discussion section. And the descriptions are not logical and distracted!

7) The authors did NOT analyze RNA but used "transcripts" to describe the gene copy abundances throughout the manuscript.

8) All figures were unclear. There are Chinese characters in Figure 2. The phylogenetic tree is too rough..

I cannot list line-by-line specific comments because of too many wrong/problems throughout the manuscript.

---

## Author Comment (AC1) · 30 Mar 2019

C1: The first issue is a lack of strong research motivations. While the Introduction briefly reviews the background about DNRA and previous work, it lacks a rational development of research questions, motivations, or working hypotheses. The authors should stress more in the Introduction on what the key unaddressed questions are after reviewing the previous work (e.g., page 2, lines 13-30; page 3, lines 3-7), why they are important, and how the results from the HSE/PRE region would help resolve these unknowns. One way to improve is to start off with a bigger picture of N cycle/unknows (rather than a description of the HSE region), followed by an overview of previous work

on DRNA that subsequently leads to major research questions, and then justification on the choice of PRE sites (not just because "there is lack of study : : : from PRE"). Note that the fact that there hasn't been any study in the PRE region is by itself not an attractive incentive because most audiences would probably not be familiar or interested in PRE or HSE region in particular, unless such environment is unique for this kind of research and can provide new insights.

R1: Thanks for the reviewer's comment. We have made a significant modification on the Introduction, to address the rational development of research questions, motivations, or working hypotheses, in the revised manuscript. See page 1,2, introduction please.

C2: The second major issue is the presentation of data. The manuscript is written in a data-reporting style with very little discussion on the data. The authors are expected to place their results in the context of the existing literature and synthesize their findings into lucid scientific story that advances our understanding of N cycle in a broader sense (i.e., not just for the PRE region). A "real" Discussion section is currently missing from the manuscript. Another issue with data presentation is that most figures are of low quality and it is very difficult to read the number, text, or color (e.g., Figs. 2, 3, 4, 5, 7,8).

R2: More discussion on the data has been provided in the revised manuscript, including 3.1 Environmental Gradients of the Estuary, 3.2 The potential rates of DNRA, 3.3 DNRA rates and nrfA gene abundance, 3.4 Spatial Distribution Patterns in the Sediment. The figures mentioned by the reviewer have been modified to improve their quality. Fig 3,5 have been remake by Arcgis soft. Fig 7 was still difficulty to show the minor number of bacteria. Fig 8 is hard to modified.

C3: The last issue is the frequent improper uses of English and language that need to be checked thoroughly throughout the manuscript. A few examples are given below.

Line 17: English. I would suggest replacing "accordingly" with "the corresponding"

throughout the manuscript. Line 25: English. Insert "in" after "located" Line 27: English. "As the marine: : :" This sentence has grammatical issues; please rephrase. Page 2 Line 2: English. "From which the good: : :". This should be a clause that follows the previous sentence; please check the grammar. Line 5: English suggestion: change "forming" to "converted to" Line 13: English. Delete "of" and "were" Page 3 Line 7: English. Remove "the" from "there are the" Line 19: English. Insert "is" before "refereed" Line 24: English. Insert "in" after "located" Page 4 Line 1: English. Replace "transported" with "transport" Line 12: "and so on". Please list all the sampling time points. Page 6 Line 12: Please specify the "sea water quality standards, GB3097-1997". Line 14: What are "the standard four level" and "80% datas"? Line 22: English. "rates significantly among". Please check the grammar. Line 23: English. "locate" should be in past tense. R3: All these mistakes have been corrected and revised C4: Line 8: The authors could elaborate the geographic and climatic differences between HSE and other study sites (ECS and YRE); that will help justify the choice of the HSE for this study. Page 7 Line 2: "17000 cl-1 mg/L". Please use the more common unit for salinity (e.g., Practical Salinity Unit) in the manuscript. Lines 3-26: The presentation is quite descriptive. Same issue is found on page 8. It's partially because the presentation of data is frequently referred to specific sites (e.g., C1, C7, C11), which have limited meanings to the general audience. Just as an example, a description of "higher XX rates were found at Stns 1, 2, 3: : :" is less interesting/ meaningful than a description of "higher XX rates were found in lower-salinity regions (such as Stn1, Stn2, Stn3)". I strongly suggest to separate the Results section from the Discussion section.

R4: All these comments have been considered and revisions have been made.
* * *
**Environment Gradient related Dissimilatory Nitrate Reduction to Ammonium in Huangmao Sea Estuary: Rates and Community Diversity**

Ran Jiang[1,2], Shu-Xin Zhang[3], Wei Guo [1,2], Rui He[1,2], Li-Ling Yang[1,2], Xin-Sheng Chai[4]

5  [1]The Pearl River Hydraulic Research Institute, Pearl River Water Resources Commission of the Ministry of Water Resources, Guangzhou, 510640, China
[2]Key Laboratory of the Pearl River Estuarine Dynamics & Associated Process,Ministry of Water Resources, Guangzhou , 510640, China
[3]School of Chemical Technology, Guangdong Industry Polytechnic, Guangzhou, Guangdong Province, 510300, China
10  [4]State Key Laboratory of Pulp and Paper Engineering, South China University of Technology, Guangzhou, 510640, China

*Correspondence to*: Ran Jiang(458750084@qq.com)

**Abstract.** The potential rates of dissimilatory nitrate reduction to ammonium (DNRA) in the sediments collected from Huangmao Sea Estuary (HSE), one of Pearl River Estuaries in China, were investigated. It was based on the $^{15}N$ isotope and molecular biology analysis at 20 sites of HSE in one-year period. The results showed that the environment gradients of

15  nitrogen pollutants decreased from the estuary to the sea, and the characteristics of terrestrial pollutant export were obvious. $^{15}NH_4^+$ was detected in all sites in continuous-flow systems, which indicated the presence of DNRA pathway in HSE. Both the highest DNRA rate and *nrfA* gene fragments were found in the site located near coastal pollution zone. While the lower DNRA rates were measured in the sediment samples from sites near the White Dolphin National Nature Reserve, which were less than 3 ug/L.h in 72 hours. Principal components analysis (PCA) based on the community composition at genus

20  level was performed , and nine samples formed three distinct clusters, corresponding to their geographical locations, which suggested that the salinity was likely to affect the selection of the *nrfA* bacterial populations. This study provides a better understanding of the DNRA in the Pearl River Estuary.

Key word: DNRA; Sediment; Rate; Distribution pattern

25  **1 Introduction**

The Pearl River Estuary (PRE) is one of the most complex estuarine systems in the world, forming a transition zone between the Pearl River and the South China Sea (Pan et al. 2014).  Many environmental issues have subsequently emerged, such as overloading of reactive nitrogen and hypoxic zones (Dai et al. 2008). Huangmao Sea Estuary (HSE) is one of the estuaries located in the western part of PRE (as shown in Fig. 1), a fastest growing economic area in the southern

30  China. According to the marine quality bulletin of Jiangmen city, the active nitrogen (including $NO_3^-$, $NH_4^+$ and $NO_2^-$) was

**Fig. 1.**

---

## Author Comment (AC2) · 30 Mar 2019

C1: The English of the paper needs an overhaul. Most of the sentences are wrong, unclear, lacking logic, or misleading. Some references cited in the main text are missing in the list.

R1: The English of the paper has been carefully checked and some paragraphs have been completely rewritten. The missing references have been added in the list.

C2: The methods were not described clearly. Many key details are missing. For example, the one-year observations were not stated; the biogeochemical parameters

(temperature, salinity, nutrient concentrations, etc.) measurements are missing; the procedure of the high-throughput sequencing of the nrfA gene is incomplete: : :

R2: In the revised manuscript, the description of the methods has been rewritten.

C3: The method on time-series incubations with supplemental 15N-NO3- to examine potential rates of DNRA needs to clarify the cross-feeding effect since heterotrophic bacteria can uptake 15N-NO3- and release 15N-NH4+ to the experiment system.

R3: We take most denitrifying and DNRA bacteria can grow heterotrophically while some grow autotrophically. In the revised manuscript, the clarification for the method on time-series incubations with supplemental 15N-NO3- to examine potential rates of DNRA has been provided.

C4: There is no statistical analysis in this manuscript. All statements and conclusions on the control of environmental parameters on communities are not supported by any specific analysis or statistical test.

R4: The statistical analysis has been provided in the revised manuscript, to support the statements and conclusions addressed in this work.

C5: The whole paragraph of "2.4 Rate calculations" (Page 4, Line 22-30) was plagiarized from the literature. Also, some statements in the Introduction section (Page 2, Line 17- 22) were from the literature directly.

R5: These sentences have been revised.

C6: Very few discussion was made in the Results and discussion section. And the descriptions are not logical and distracted!

R6: More discussion has been provided in the "Results and discussion section". We paid more attention to present them logically.

C7: The authors did NOT analyze RNA but used "transcripts" to describe the gene copy abundances throughout the manuscript.

R7: The mistake has been corrected.

C8: All figures were unclear. There are Chinese characters in Figure 2. The phylogenetic tree is too rough.

R8: Some figures are revised. The mistake in Fig. 2 has been corrected.

Please also note the supplement to this comment:
https://www.biogeosciences-discuss.net/bg-2019-31/bg-2019-31-AC2-supplement.pdf

―――――――――――――――――――

**Supplement:**

**Environment Gradient related Dissimilatory Nitrate Reduction to Ammonium in Huangmao Sea Estuary: Rates and Community Diversity**

Ran Jiang[1,2], Shu-Xin Zhang[3], Wei Guo [1,2], Rui He[1,2], Li-Ling Yang[1,2], Xin-Sheng Chai[4]

[1]The Pearl River Hydraulic Research Institute, Pearl River Water Resources Commission of the Ministry of Water Resources, Guangzhou, 510640, China
[2]Key Laboratory of the Pearl River Estuarine Dynamics & Associated Process,Ministry of Water Resources, Guangzhou , 510640, China
[3]School of Chemical Technology, Guangdong Industry Polytechnic, Guangzhou, Guangdong Province, 510300, China
[4]State Key Laboratory of Pulp and Paper Engineering, South China University of Technology, Guangzhou, 510640, China

*Correspondence to*: Ran Jiang(458750084@qq.com)

**Abstract.** The potential rates of dissimilatory nitrate reduction to ammonium (DNRA) in the sediments collected from Huangmao Sea Estuary (HSE), one of Pearl River Estuaries in China, were investigated. It was based on the $^{15}$N isotope and molecular biology analysis at 20 sites of HSE in one-year period. The results showed that the environment gradients of nitrogen pollutants decreased from the estuary to the sea, and the characteristics of terrestrial pollutant export were obvious. $^{15}NH_4^+$ was detected in all sites in continuous-flow systems, which indicated the presence of DNRA pathway in HSE. Both the highest DNRA rate and *nrfA* gene fragments were found in the site located near coastal pollution zone. While the lower DNRA rates were measured in the sediment samples from sites near the White Dolphin National Nature Reserve, which were less than 3 ug/L.h in 72 hours. Principal components analysis (PCA) based on the community composition at genus level was performed , and nine samples formed three distinct clusters, corresponding to their geographical locations, which suggested that the salinity was likely to affect the selection of the *nrfA* bacterial populations. This study provides a better understanding of the DNRA in the Pearl River Estuary.

Key word: DNRA; Sediment; Rate; Distribution pattern

**1 Introduction**

The Pearl River Estuary (PRE) is one of the most complex estuarine systems in the world, forming a transition zone between the Pearl River and the South China Sea (Pan et al. 2014).  Many environmental issues have subsequently emerged, such as overloading of reactive nitrogen and hypoxic zones (Dai et al. 2008). Huangmao Sea Estuary (HSE) is one of the estuaries located in the western part of PRE (as shown in Fig. 1), a fastest growing economic area in the southern China. According to the marine quality bulletin of Jiangmen city, the active nitrogen (including $NO_3^-$, $NH_4^+$ and $NO_2^-$) was

one of the major eutrophication pollutants in HSE (Jiangmen Marine and Fisheries Bureau, 2016). Since the nitrogen related eutrophication has become a severe problem in the Pearl River Delta area, it is important to have a further understanding of nitrogen cycles, i.e., denitrification and nitrification in this area. From which the good strategy for the eutrophication control and ecological restoration can be made.

5    There are several ways to remove inorganic nitrogen from the ecosystem in estuary (Herbert, 1999). Denitrification is regarded as dominant process for the nitrate reduction in the shallow marine sediments, in which nitrate was eventually converted to nitrogen and released from water body (Herbert, 1999). Unlike denitrification and anammox, the transformation from nitrate to ammonium via dissimilatory nitrate reduction to ammonium (DNRA) could prolong the residence time of fixed nitrogen and thus has a negative impact for removing the bioavailable nitrogen from the ecosystem (Megonigal et al., 10   2004). At present, most studies on the DNRA pathways relied on the correlations between the DNRA rates and environmental factors, such as salinity, temperature, concentrations of $NO_3^-$, organic matter. For example, Laverman et al.'s work showed that the increase of salinity (by 10 psu) may, at least temporarily, increase the efflux of $NH_4^+$ from the sediment to the surface water, which limits the ability of denitrification to remove bioavailable nitrogen from aquatic ecosystems (Laverman et al., 2007). It is often thought to occur when $NO_3^-$ inputs exceed the availability of carbon substrate 15   for denitrification (Tjedje, 1998). In Gardner and McCarthy's work (Gardner and McCarthy, 2009), they found that the high salinities (31.4 - 45.1 psu), temperature, and organic carbon availability may exacerbate the effects of nutrient inputs by maintaining the bioavailable N in the system as the form of $NH_4^+$ or $NO_2^-$ via DNRA or DNRN (dissimilatory $NO_3^-$ reduction to $NO_2^-$ ), rather than having it to be removed via denitrification or anammox. Hardison at el.'s work (Hardison at el. 2015) indicated that high organic C loading is a prerequisite for DNRA to be favour over the denitrification, but the $N_2$ 20   production may still be significant when organic C is highly depending on $NO_3^-$ availability. However, the direct assessment of the microbial communities inhabiting in changing salinity environment as saltwater-fresh water interface moved to inland is limited. Further, as the potential microbial responses involving the N transformations to the changing salinity, the microbial communities that promote DNRA in HSE is necessary to be studied.

   There are some studies available on the organisms capable of DNRA in the coastal sedimentary environments (Gardner et al., 25   2009, Gardner et al.,2006), although it is difficult to determine the exact organisms involved. The functional gene assay (Smith et al., 2003, Takeuchi, 2006), i.e., using a nitrate ammonification related gene -- *nrfA* (named after nitrite reduction by ferment, coding for another dissimilatory nitrite reductase containing cytochrome c552 [EC 1.7.2.2]), first developed from anaerobic wastewater treatment reactors in 2004 (Mohan, et al. 2004), could be an effective approach to be used in such investigations. By checking the distribution and diversity of *nrfA* gene encoding DNRA in the sediments, it was found that 30   most of the *nrfA* sequences from the estuarine sediments of Clone River (UK) have several distinct clusters, including known nitrate ammonifiers, e.g., *Aeromonas*, *Shewanella, Desulfovibrio* and *Sulfurospillum* (Takeuchi, 2006). This is also the first trial to describe the *nrfA* partial sequences derived from a natural environment, with reference to their habitat-specific

community structure (Mohan, et al. 2004). However, most of the studies mentioned above focused only on the *nrfA* gene encoding DNRA, they paid less attention on the observable DNRA rate so far.

Some investigations have been also conducted for the sediment from different coastal areas in China. For example, the study showed that the N-loss as $N_2$ was the main pathway in East China Sea (ECS), in where DNRA counts for 20–31% of benthic
nitrate reduction (Song et al., 2013). The investigation on the sediments from Yellow River Estuary (YRE) indicated that variability in the activities and community structure of DNRA bacteria (Cuina et al., 2017). There were several reports for showing the coexistence of denitrification and DNRA in Yellow River and Changjiang River in ECS. However, there are significant differences in both geographic location and climatic conditions between PRE and ECS or YRE. The PRE and ECS, as two largest estuaries in China in terms of freshwater discharge, are located in subtropical and temperate region,
respectively (Liu et al., 2018). YRE consists of typical turbid coastal waters that are influenced by the Yellow River which is the second largest river in the world in terms of sediment loadings, is located in monsoon and temperate region (Cui et al., 2011). Hong et al.'s work (Hong et al., 2019) indicated that estuaries play an important ecological role in the removal of overloaded nitrogen, thus relieving eutrophic pressure in coastal areas in PRE. To date there are lack of studies on the microbial mediated nitrate reduction pathways, especially DNRA that causes the active nitrogen delay in the PRE ecosystem.
Therefore, it is necessary to conduct a genetic investigation based on the sediments collected from PRE areas, in order to have a better understanding of DNRA pathway in PRE.

In this work, we chose HSE, one of the PRE, as the model area in the present investigation. The major focuses were to (1) examine the potential rates of DNRA in the sediments collected from the HSE continental shelf; (2) to develop the *nrf*A specific primer pairs for describing the distribution and phylogenetic status of the nitrate ammonifiers inhabited in the
sediments; and (3) to figure out the impact of the salinity and overloaded nitrogen on the DNRA. From which not only the environmental-gradient related DNRA profile would be revealed but also the activity and community compositions for the DNRA bacteria in these sediments were identified.

**2 Materials and methods**

**2.1 Details of the area selected**

HSE (see Fig. 1) is referred to as Huangmao Bay with an area of approximately 540 km$^2$. It is composed of a bay proper in the lower portion, a tidal river in the upstream and several island chains (Wei and Wu, 2011). The average water depth in the Bay is approximately 4.5 m. The Bay width changes from 35 km at the mouth to 1.9 km at the head, forming a funnel-shaped estuary (Wei and Wu, 2001). The Bay mouth features a chain of rock islands (Gaolan, Hebao and Dajing Islands) that provides a shelter from wave attack, resulting in an ideal place for port construction and tourism development. There
exists a deep navigation channel extending from the Bay mouth towards the Bay head, with water depths ranging from 6 to 22 m. The studied sites were located in the Bay mouth, around Dajing Islands. Sites C16-20 were near the west boundary of

the PRE Chinese White Dolphin National Nature Reserve. C3 was in the scope of influence of warm drainage pige and domestic sewage discharge from an factory.

**2.2 Sample collection and preparation**

Nine (9) sediments samples located in the sites C1, C3, C5, C8, C11, C13, C15, C16, and C17 were collected at twenty sites (Fig. 1) during a cruise from Jun 8 to 22, 2016. These sediments samples were used for measuring the potential DNRA rate. Each sampling series involved taking three sediment cores from every station. A detailed description of these sites were listed in Table 1. Undistrubed sediment cores were collected by means of plexiglass tubes, driven directly into the sediment. After closing the tubes with a rubber stopper at the bottom, they were placed in a thermostatic container (which was enabled to preserve natural conditions) and transported from the sampling sites to the laboratory. During transportation, the container was filled with a 20-centimeter layer of overlaying water from the sediment, as the source of oxygen and nutrients (An, et al. 2002).

At laboratory, each intact core was installed into a continuous-flow system (Fig. 2) consisting of an intake water vessel, flow tubes, intact sediment core, peristaltic pump, incubation bath, and sample collection vessel (Wayne, et al., 1995). A plunger, with Teflon inlet and outlet tubes and sealed to the core cylinder with an o-ring, was installed over each sediment core and adjusted to 5 cm above the sediment surface to leave an overlying water volume (~ 230 mL). Each core was incubated at near in situ temperature and under laboratory light conditions, and site bottom water was passed continuously over the core surface with a flowrate of 1.5 mL min$^{-1}$. After 24-h, the systems was allowed to approach steady-state conditions, and the inflow and outflow samples were collected daily for analysing the contents of dissolved gas and nutrient. After spiked 50 mg of K$^{15}$NO$_3^-$ into theflow water, the inflow and outflow water were sampled for the gas and nutrient analyses at 12, 24, 48, 72, 96, 120, 144, 168 and 216 h, respectively.

**2.3 Chemical analysis**

Concentrations of $^{28}$N$_2$ , $^{29}$N$_2$ , $^{30}$N$_2$ and Ar were measured with GasBench-isotope ratio mass spectrometry (GasBench-IRMS, CTCAnalyticsAG, Zwingen, Switzerland), which had analytical capabilities of both dual-inlet and continuous-flow modes coupled with an Agilent 6890 GC equipped with an autosampler (CTC Analytics CombiPAL), to differentiate denitrification of internal sources of N versus that derived from the added $^{15}$NO$_3^-$ in the overflowing water. The concentrations of $^{29}$N$_2$ and $^{30}$N$_2$ were calculated according to Holtappels et al.' work (Holtappels, et al., 2011). Total NH$_4^+$ and $^{15}$NH$_4^+$ (potential DNRA) production rates were measured by HPLC (Wayne, et al., 1995).

Environmental parameters including temperature, pH and salinity were on-site measured by a pH metre (PhS-3C, Rex Electric Chemical, China) and a salinometer (5150, San-xin Instrument, China). The concentrations of NO$_3^-$, DIN, NH$_4^+$ , active P in water and C$_{org}$, P, N in the sediment were determined using an ultraviolet-visible spectrophotometer (8453 UV-VIS, Agilent, USA).

**2.4 Rate calculations**

The denitrification rates of the added $^{15}NO_3^-$ (D15) and *in situ* $^{14}NO_3^-$ (D14) were calculated from the measured production of $^{14}N^{15}N$ ($p^{29}N_2$) and $^{15}N^{15}N$ ($p^{30}N_2$): D15= $p^{29}N_2 + 2p^{30}N_2$ ; D14 = ($p^{29}N_2$ /2 $p^{30}N_2$ )·D15 (Gruca-Rokosz, et al.,2009). The total *in situ* rate of dissimilatory $^{14}NO_3^-$ reduction to $NH_4^+$ (DNRA) was estimated from the production rate of

5    $^{15}NH_4^+$ ($p^{15}NH_4^+$) in an anoxic–anaerobic $NO_3^-$ reduction zone. Assuming that DNRA takes place in the same sediment stratum as denitrification, the $^{15}N$ atom% of $NO_3^-$ reduced to $NH_4^+$ is the same as the $^{15}N$ atom% of $NO_3^-$ reduced to $N_2$. The in situ rate of DNRA can therefore be calculated as: DNRA = $p^{15}NH_4^+$·(D14/D15) (Gruca-Rokosz, et al.,2009, Giblin, et al., 2010). Further details are given by Risgaard–Petersen and Rysgaard (Risgaard-Petersen, and Rysgaard, 1995). The partitioning of DNRA and $N_2$ production was expressed as %DNRA, i.e., = 100[DNRA/(DNRA + $N_2$ production)], with total

10    $N_2$ production in the model or $p^{30}N_2$ and $^{29}N_2$ in the vials accounting for the remainder (Hardison, et al. 2015). The significance of the rate differences in the continuous-flow system were evaluated by the t-test or the Mann–Whitney Rank Sum Test, using a statistical software package (SigmaStat 3.5).

**2.5 Molecular biology analysis and DNRA communities**

**DNA Extraction from the sediments.** Bulk sediment samples were collected in the sites C1, C3, C5, C8, C11, C13, C15,

15    C16 and C17 from upper (0 – 3 cm) layers using an ethanol washed spatula, and stored in the box filled with crashed ice. DNA was extracted from wet sediment samples (1.0 g) using the PowerSoil DNA Kit (MO BIO Laboratories, USA) according to manufacturer's instructions, and the concentrations were measured on a spectrophotometer (E-Gel Imager, U.S.).

**Design of PCR primers.** Primer pairs (6F/5R and 6F/4R) were designed based on the existing partial *nrf*A sequences and

20    complete sequences including the nrfA fragment in an open reading frame (ORF). New pairs of forward and reverse primers (6F/5R) were manually designed based on the guidelines (Alkami Biosystems 1999) and the ClustalW (Thompson et al. 1994) multiple DNA sequence alignment. Two conserved regions of the nrfA sequences, consisting of one based on six sequences (AE008900, AL627282, AE005640, X72298, U32787, AE006035) and the other based on four sequences (AE005640, X72298, AE008900, AL627282) were chosen as the target of designing the new primer pairs, 6F/5R,

25    respectively. Another primer set (6F/4R) developed by Takeuchi (2016) was also employed after the PCR condition was optimized.

Sequences of the nrfA primers used in this study are listed in Table 2. Compared with Primer pairs (6F/4R), the PCR produces of primer pairs (6F/5R) had more conservative sequences for the high-throughput sequencing of nrfA gene in HSE Estuary samples.

30    **High-throughput sequence of *nrf*A gene .** The DNRA community diversity of DNRA bacteria was determined on an Illumina MiSeq sequencer. Sites C1, C3, C5, C8, C11, C13, C15, C16 and C17 were selected to examine the community structure. After DNA extraction, PCR was conducted in triplicate using PCR Amplifier 2720 under the same conditions as

those reported previously (Song, et al. 2014). The PCR amplification was performed using a programmable thermocycler, Gene Amp 9700 (Applied Biosystems). The programmers consisted of the following conditions: the initial denaturation at 94∘C for 2 min, 30 cycles of 58°C for the first-round PCR (20 cycles for the second-round PCR) for 1 min, followed by an annealing temperature at 46.5∘C for the first-round PCR and at 54.5∘C for the second-round PCR.

5    **Sequence analysis and phylogenetic classification.** PCR result was quantified using a PicoGreen dsDNA Assay Kit (Termo Scientifc, USA) on the TBS-380 Mini-Fluorometer (Turner Biosystems, CA, USA). Then, paired-end pyrosequencing was conducted using MiSeqReagent Kit V3 (Life Sciences, Branford, CT, USA) by the Illumina MiSeq sequence platform at the RayBiotech (Guangzhou, China). Raw reads of *nrf*A gene sequences were saved in the FsatQ file. First, the sequences shorter than 150 bp and lower than 20 quality score, as well as the sequences containing ambiguous base

10    N, were removed through the sliding window method. Paired reads were merged based on the overlap of bases using FLASH (http://ccb.jhu.edu/software/FLASH/) (Magoc., 2011). Then, the valid sequences were obtained through the identification between the merged reads and the relevant samples. Chimera sequences were removed using QIIME (http://qiime.org/) (Edgar, 2010) to get the high quality sequences, which were clustered into OTUs using UCLUST56 with 97% similarity and rare OTUs containing lower than 0.001% of total *nrfA* sequences were omitted (Edgar, 2011). The sequences with highest

15    abundance of each OTU as representative sequences were annotated with NCBI taxonomy using QIIME (soft version 1.9.1). Microbial community richness indices included Chao1 estimations, while diversity was quantified using the Simpson and Shannon indexes (Edgar, 2011). The alpha diversity index of each sample was determined using QIIME after a flattening process sequence (Page, 1996). Rank abundance curve was drawn using R software. The length of X-axis represented richness and slope of lines represented evenness. A steep gradient indicated low evenness. The differences of community

20    composition among four sites along the salinity gradient were evaluated through PCA based on community composition at genus level using QIIME software. Distributions of DNRA bacteria in four samples at different levels were determined using QIIME, and a heat map was constructed with the 50 most abundant genera using R software. In addition, RDA was conducted to evaluate the relationship between environmental factors and community structures using R software. Phylogenetic trees were generated using the neighbour-joining method as implemented with a TreeView programme (Page,

25    1996).

**3 Results and discussion**

**3.1 Environmental Gradients of the Estuary**

Bottom seawater and sediment characteristics were investigated at 20 stations (Fig. 1). Twenty (20) stations in the sites C1-C20 had been sampling quarterly for a one-year period during 2016. The national sea water quality standard in China (NO.

30    GB3097-1997), approved by the Ministry of Environmental Protection, was used in assessing the sea water quality of HSE. The Standards classified the seawater quality into four categories, and each grade of water is served for different purposes. The fourth grade of sea water are suitable for the marine development zones. The standard dissolved inorganic nitrogen

(DIN) is referred to the totality of $NO_3^-$, $NO_2^-$ and $NH_4^+$. It showed that 80% data of $NO_3^-$, DIN and $NH_4^+$ in the sites C1-C17 exceeded the four grade. While the areas in the sites C18-20, the water had better quality, which reached at the third grade. The sites C18-20 are located in near the PRE Chinese White Dolphin National Nature Reserve, and land pollutants were not so easy to spread in the area. As showed in Fig. 3a, DIN decreased sharply from the estuary coast ($\sim$ 0.842 mg/L) to the outer shelf ($\sim$0.270 mg/L). In the same time, $NO_3^-$ range in 0.17 - 0.567mg/L, which accounted for 60 - 80% of DIN. The concentrations of DIN in about 80% sites excessed 0.5 mg/L. The maximum concentration of DIN appeared at the mouth of Yamen Gate, decreases from upstream and estuarine coast to downstream. In general, the concentration of DIN is higher in the west and lower in the east, as shown in the profile figure. The highest concentration in the upper estuary area was above 0.95 mg/L, while the NID near the western estuary was above 0.8 mg/L and it declined from estuary to outer shelf. Fig. 3b showed that the major form of DIN is $NO_3^-$, indicating that the pollutants in the western estuary was more serious than that of eastern area due to the hydrodynamic force. Figure 3c shows the rising trend in the concentration of $NH_4^+$, except for those observed in the sites C19-20. According to the monitoring from one-year period, the concentrations of $NO_3^-$, DIN and $NH_4^+$ were lower during the spring than summer and winter. As was shown in Figure 4, the salinity was higher in the east than the west, and it is inclined to the west, especially in flood season. The result of salinity characteristics was consistent of the hydrodynamic characteristics of the eastern tidal current in HSE. In flood season, the salinity was affected by fresh water runoff, so it was less than in the dry period. The salinity in our studied area was near the salinity contour (15000-17000 cl$^-$ mg/L) in bottom water.

**3.2 The potential rates of DNRA**

$^{15}NH_4^+$ was detected in all sites after 12h in continuous-flow systems, which showed that DNRA existed in continuous-flow system. $^{15}NH_4^+$ ranged from 310.0 to 1948.5 ug/L after 216 h's experiments. The in situ potential rate of DNRA was different in every sampling sites. Moreover, the maximum $^{15}NH_4^+$ was 1948.5 ug/L in the site C3, the corresponding in situ rate of DNRA was 23.3 ug/(L.h) in 25h, and 6.27ug/(L.h) in the end of 216 h. The second maximum $^{15}NH_4^+$ was 1703.1 ug/L in the site C17, the corresponding in situ rate of DNRA was 2.75 ug/(L.h). While the low $^{15}NH_4^+$ was 310.0 ug/L, 398.2 ug/L in the sites C8 and C16, accordingly in situ rate of DNRA was 1.53 and 2.36 ug/(L.h). The in situ rate of DNRA increases with the growth of cultivation time and reaches at the stable in five days, in which peak value was about 23.3 ug/(L.h), in one to three days. After five days, the rate was ten slowed down to the range of 0.28 - 13.4 ug/(L.h).

Sample sites have different hydrodynamic and pollutant characteristics, the sites C1 and C11 were in the upstream runoff channel, exposing to upstream pollutants, while the sites C5 and C7 were relatively near the coastline and the site C3 was the nearest these domestic sewage pipes and warm drainage (below sea level). As were shown in Figs. 5a and 5b, the potential DNRA rates in the spatial distribution prove the insignificant statistical differences between the particular stations in salinity range, indicating the high potential DNRA rate near the sewage discharge in western coastal. It is believed that the most important factors effected DNRA were related to the real environment of the functional microorganisms. Furthermore, the

temperature, salinity, nitrogen and phosphorus may also be the important factors regulating the fate of $NO_3^-$. Due to the role of DNRA in nitrogen conversion, it could be included in the active nitrogen pollution pathway in the southern China.

**3.3 DNRA rates and nrfA gene abundance**

The abundances of DNRA bacteria based on *nrfA* gene quantification in the sediments of 9 sampling sites in Supplementary Table S1. The two *nrfA* primer (6F/4R) and (6F/5R) were used in high-throughput sequencing respectively. Effective tags of 6F/5R *nrfA* primer are in the range of $11 \times 10^6$ to $7 \times 10^5$, while that of 6F/4R *nrfA* primer is in range of $7 \times 10^5$ to 9. Therefore, the analysis of DNRA community should be based on 6F/5R *nrfA* primer. High contents of *nrfA* gene fragments were found at the site C3, with values of 1159715 copics/(g wet sediment). Correspondingly, very low contents of *nrfA* were found at the sites C8, C16 and C15.

The potential DNRA rates was ranged from 0.13 ug/L.h to 23.3 ug/L.h in 72 hours. The highest DNRA rate was found in the site C3, which is located near coastal pollution zone. The lower rates were found in the sediment sampled at the sites C16 and C18, which is located near the White Dolphin National Nature Reserve. The sites C16 and C18 have lower DNRA rates, less than 3 ug/L.h in 72 hours. The maximum $^{15}NH_4^+$ was 1948.5ug/L in the site C3, the corresponding in situ rate of DNRA was highest in nine sites. Thus, potential DNRA rate and *nrfA* gene fragments had a positive correlation.

[revised manuscript text omitted]
, based on the 20 sites of HSE during a one-year period. The results showed that the maximum $^{15}NH_4^+$ was 1948.5 ug/L, the corresponding in situ rate of DNRA was 6.3ug/L.h in 72h. High contents of *nrfA* gene fragments were found at the same site, with values of 1159715 copics/(g wet sediment). Correspondingly, very low contents of *nrfA* were found at other sites. The distribution pattern of the *nrf*A gene sequences from the surface sediments formed three distinct clusters corresponding to their geographical locations. which suggests that salinity per se was likely to affect the selection of the *nrfA* bacterial populations. Base on the nrfA gene pyrosequencing, *Enterobacteriaceae* (a member of *Enterobacteriales*, were more abundant than other families) counts for 89.9 - 99.0% of the classified sequences. This study improves our understanding of DNRA in the PRE. However, the further study is required to understand the contributions of anammox, denitrification and DNRA to nitrate reduction.

**Acknowledgments**

This study is funded by the National Natural Science Foundation of China (Grant no. 51409287) and the Guangdong Province Natural Science Foundation of China (Grant no. 2017A030313329).

**Table1. Water and sediment data in HSE**

| Site ID | Salinity psu | pH | NH$_4^+$ | Active P ug/L | NO$_3^-$ | Corg % | N mg/kg | P |
|---|---|---|---|---|---|---|---|---|
| | | | Surface water (0-50cm) | | | Sediment (0-5cm) | | |
| C1 | 0.3 | 7.76 | 1.3 | 42 | 633 | 1.70 | 1017 | 867 |
| C3 | 1.0 | 7.96 | 1.9 | 32 | 622 | 1.25 | 896 | 835 |
| C5 | 12.9 | 7.98 | 2.2 | 28 | 532 | 0.71 | 516 | 468 |
| C8 | 13.4 | 8.12 | 3.3 | 29 | 504 | 1.00 | 596 | 729 |
| C11 | 2.15 | 7.74 | 0.9 | 32 | 495 | 1.37 | 897 | 732 |
| C13 | 9.55 | 7.61 | 0.9 | 35 | 475 | 1.00 | 867 | 839 |
| C15 | 12.5 | 7.53 | 0.6 | 34 | 494 | 1.38 | 861 | 523 |
| C16 | 9.9 | 7.66 | 1.1 | 40 | 415 | 1.24 | 723 | 589 |
| C17 | 13.0 | 7.72 | 1.9 | 29 | 418 | 0.98 | 689 | 368 |

Temperture in the area: 22-26 $^o$C

25  **Table 2. Sequences of the nrfA primers used in this study**

| Primer | Sequence(5′ to 3′) |
|--------|--------------------|
| 5R | CGCCAYTGVGCRTGRCGRATATC |
| 6F | GAYTGCCAYATGCCRAAAGT |
| 4R | GCATCCGCSGCTTTATCCAT |

F: forward primer, R: reverse primer;

Base Codes: K=T/G, R=A/G, Y=C/T, S=C/G

[Figure]

[Figure]

**Figure 1. Location of sampling stations in HSE**

[Figure]

**Figure 2. Diagram of Continuous-flow System**

[Figure]

**Figure 3a. DIN Spatial distribution**

[Figure]

**Figure 3b. NO$_3^-$ Spatial distribution**

[Figure]

**Figure 3c. NH$_4^+$ Spatial distribution**

[Figure]

**Figure 4. Flat Distribution of Chlorine Content in the bottom of HSE at maximum tide level, unite psu**

[Figure]

**Figure 5a. Potential DNRA Rates Spatial distribution in 24 hours**

[Figure]

5    **Figure 5b. Potential DNRA Rates Spatial distribution in 48 hours**

[Figure]

**Figure 6a. PCA analysis of DNRA communities in HSE. Unifrac phylogenetic distance calculation method based on Quantitative Weighted was used to calculate clustering. Percentages represent the amount variance explained by each dimension**

[Figure]

**Figure 6b. Unweighted Pair Group Method with Arithmetic Mean(UPGMA) analysis of DNRA communities in HSE**

[Figure]

**Figure 7a. Distributions of DNRA bacteria in HSE at the phylum level (b)the class level (c) the order level (d) the family level**

[Figure]

5  **Figure 7b. Distributions of DNRA bacteria in HSE at the class level**

[Figure]

**Figure 7c. Distributions of DNRA bacteria in HSE at the order level**

[Figure]

5    **Figure 7d. Distributions of DNRA bacteria in HSE at the family level**

[Figure]

**Figure 8. Dendrogram showing the diversity of the *nrf*A partial nucleotide from the sediments in HSE**, with reference sequences from the databases: *Escherichia coli, Shewanella, Sulfurospirillum, Dusulfovibrio* and so on. The bar indicates 10% nucleotidesequence divergence. Numbers near the branches represent the percentages of 1,000 bootstrap repetitions. Confidence limits of less than 50% are not shown.

---

## Author Comment (AC3) · 30 Mar 2019

C1: The first issue is a lack of strong research motivations. While the Introduction briefly reviews the background about DNRA and previous work, it lacks a rational development of research questions, motivations, or working hypotheses. The authors should stress more in the Introduction on what the key unaddressed questions are after reviewing the previous work (e.g., page 2, lines 13-30; page 3, lines 3-7), why they are important, and how the results from the HSE/PRE region would help resolve these unknowns. One way to improve is to start off with a bigger picture of N cycle/unknowns (rather than a description of the HSE region), followed by an overview of previous work

on DRNA that subsequently leads to major research questions, and then justification on the choice of PRE sites (not just because "there is lack of study : : : from PRE"). Note that the fact that there hasn't been any study in the PRE region is by itself not an attractive incentive because most audiences would probably not be familiar or interested in PRE or HSE region in particular, unless such environment is unique for this kind of research and can provide new insights.

R1: Thanks for the reviewer's comment. We have made a significant modification on the Introduction, to address the rational development of research questions, motivations, or working hypotheses, in the revised manuscript. See page 1,2, introduction please.

C2: The second major issue is the presentation of data. The manuscript is written in a data-reporting style with very little discussion on the data. The authors are expected to place their results in the context of the existing literature and synthesize their findings into lucid scientific story that advances our understanding of N cycle in a broader sense (i.e., not just for the PRE region). A "real" Discussion section is currently missing from the manuscript. Another issue with data presentation is that most figures are of low quality and it is very difficult to read the number, text, or color (e.g., Figs. 2, 3, 4, 5, 7,8).

R2: More discussion on the data has been provided in the revised manuscript, including 3.1 Environmental Gradients of the Estuary, 3.2 The potential rates of DNRA, 3.3 DNRA rates and nrfA gene abundance, 3.4 Spatial Distribution Patterns in the Sediment. The figures mentioned by the reviewer have been modified to improve their quality. Fig 3,5 have been remake by Arcgis soft. Fig 7 was still difficulty to show the minor number of bacteria. Fig 8 is hard to modified.

C3: The last issue is the frequent improper uses of English and language that need to be checked thoroughly throughout the manuscript. A few examples are given below.

Line 17: English. I would suggest replacing "accordingly" with "the corresponding"

throughout the manuscript. Line 25: English. Insert "in" after "located" Line 27: English. "As the marine: : :" This sentence has grammatical issues; please rephrase. Page 2 Line 2: English. "From which the good: : :". This should be a clause that follows the previous sentence; please check the grammar. Line 5: English suggestion: change "forming" to "converted to" Line 13: English. Delete "of" and "were" Page 3 Line 7: English. Remove "the" from "there are the" Line 19: English. Insert "is" before "refereed" Line 24: English. Insert "in" after "located" Page 4 Line 1: English. Replace "transported" with "transport" Line 12: "and so on". Please list all the sampling time points. Page 6 Line 12: Please specify the "sea water quality standards, GB3097-1997". Line 14: What are "the standard four level" and "80% datas"? Line 22: English. "rates significantly among". Please check the grammar. Line 23: English. "locate" should be in past tense. R3: All these mistakes have been corrected and revised C4: Line 8: The authors could elaborate the geographic and climatic differences between HSE and other study sites (ECS and YRE); that will help justify the choice of the HSE for this study. Page 7 Line 2: "17000 cl-1 mg/L". Please use the more common unit for salinity (e.g., Practical Salinity Unit) in the manuscript. Lines 3-26: The presentation is quite descriptive. Same issue is found on page 8. It's partially because the presentation of data is frequently referred to specific sites (e.g., C1, C7, C11), which have limited meanings to the general audience. Just as an example, a description of "higher XX rates were found at Stns 1, 2, 3: : :" is less interesting/ meaningful than a description of "higher XX rates were found in lower-salinity regions (such as Stn1, Stn2, Stn3)". I strongly suggest to separate the Results section from the Discussion section.

R4: All these comments have been considered and revisions have been made.

Please also note the supplement to this comment:
https://www.biogeosciences-discuss.net/bg-2019-31/bg-2019-31-AC3-supplement.pdf

[Figure]

**Supplement:**

**Environment Gradient related Dissimilatory Nitrate Reduction to Ammonium in Huangmao Sea Estuary: Rates and Community Diversity**

Ran Jiang1,2, Shu-Xin Zhang3, Wei Guo1,2, Rui He1,2, Li-Ling Yang1,2, Xin-Sheng Chai4

1The Pearl River Hydraulic Research Institute, Pearl River Water Resources Commission of the Ministry of Water Resources, Guangzhou, 510640, China
 2Key Laboratory of the Pearl River Estuarine Dynamics & Associated Process, Ministry of Water Resources, Guangzhou, 510640, China

3School of Chemical Technology, Guangdong Industry Polytechnic, Guangzhou, Guangdong Province, 510300, China

4State Key Laboratory of Pulp and Paper Engineering, South China University of Technology, Guangzhou, 510640, China

Correspondence to: Ran Jiang(458750084@qq.com)

Abstract. The potential rates of dissimilatory nitrate reduction to ammonium (DNRA) in the sediments collected from Huangmao Sea Estuary (HSE), one of Pearl River Estuaries in China, were investigated. It was based on the 15N isotope and molecular biology analysis at 20 sites of HSE in one-year period. The results showed that the environment gradients of

- 15 nitrogen pollutants decreased from the estuary to the sea, and the characteristics of terrestrial pollutant export were obvious.
  15NH+4 was detected in all sites in continuous-flow systems, which indicated the presence of DNRA pathway in HSE. Both the highest DNRA rate and *nrfA* gene fragments were found in the site located near coastal pollution zone. While the lower DNRA rates were measured in the sediment samples from sites near the White Dolphin National Nature Reserve, which were less than 3 ug/L.h in 72 hours. Principal components analysis (PCA) based on the community composition at genus
- 20 level was performed, and nine samples formed three distinct clusters, corresponding to their geographical locations, which suggested that the salinity was likely to affect the selection of the *nrfA* bacterial populations. This study provides a better understanding of the DNRA in the Pearl River Estuary.

Key word: DNRA; Sediment; Rate; Distribution pattern

**25 1 Introduction**

The Pearl River Estuary (PRE) is one of the most complex estuarine systems in the world, forming a transition zone between the Pearl River and the South China Sea (Pan et al. 2014). Many environmental issues have subsequently emerged, such as overloading of reactive nitrogen and hypoxic zones (Dai et al. 2008). Huangmao Sea Estuary (HSE) is one of the estuaries located in the western part of PRE (as shown in Fig. 1), a fastest growing economic area in the southern

30 China. According to the marine quality bulletin of Jiangmen city, the active nitrogen (including  $NO_3^-$ ,  $NH_4^+$  and  $NO_2^-$ ) was

one of the major eutrophication pollutants in HSE (Jiangmen Marine and Fisheries Bureau, 2016). Since the nitrogen related eutrophication has become a severe problem in the Pearl River Delta area, it is important to have a further understanding of nitrogen cycles, i.e., denitrification and nitrification in this area. From which the good strategy for the eutrophication control and ecological restoration can be made.

- 5 There are several ways to remove inorganic nitrogen from the ecosystem in estuary (Herbert, 1999). Denitrification is regarded as dominant process for the nitrate reduction in the shallow marine sediments, in which nitrate was eventually converted to nitrogen and released from water body (Herbert, 1999). Unlike denitrification and anammox, the transformation from nitrate to ammonium via dissimilatory nitrate reduction to ammonium (DNRA) could prolong the residence time of fixed nitrogen and thus has a negative impact for removing the bioavailable nitrogen from the ecosystem (Megonigal et al.,
- 10 2004). At present, most studies on the DNRA pathways relied on the correlations between the DNRA rates and environmental factors, such as salinity, temperature, concentrations of  $NO_3^-$ , organic matter. For example, Laverman et al.'s work showed that the increase of salinity (by 10 psu) may, at least temporarily, increase the efflux of  $NH_4^+$  from the sediment to the surface water, which limits the ability of denitrification to remove bioavailable nitrogen from aquatic ecosystems (Laverman et al., 2007). It is often thought to occur when  $NO_3^-$  inputs exceed the availability of carbon substrate
- for denitrification (Tjedje, 1998). In Gardner and McCarthy's work (Gardner and McCarthy, 2009), they found that the high salinities (31.4 - 45.1 psu), temperature, and organic carbon availability may exacerbate the effects of nutrient inputs by maintaining the bioavailable N in the system as the form of  $NH_4^+$  or  $NO_2^-$  via DNRA or DNRN (dissimilatory  $NO_3^$ reduction to  $NO_2^-$ ), rather than having it to be removed via denitrification or anammox. Hardison at el.'s work (Hardison at el. 2015) indicated that high organic C loading is a prerequisite for DNRA to be favour over the denitrification, but the  $N_2$
- 20 production may still be significant when organic C is highly depending on  $NO_3^-$  availability. However, the direct assessment of the microbial communities inhabiting in changing salinity environment as saltwater-fresh water interface moved to inland is limited. Further, as the potential microbial responses involving the N transformations to the changing salinity, the microbial communities that promote DNRA in HSE is necessary to be studied.

There are some studies available on the organisms capable of DNRA in the coastal sedimentary environments (Gardner et al.,

- 25 2009, Gardner et al.,2006), although it is difficult to determine the exact organisms involved. The functional gene assay (Smith et al., 2003, Takeuchi, 2006), i.e., using a nitrate ammonification related gene -- *nrfA* (named after nitrite reduction by ferment, coding for another dissimilatory nitrite reductase containing cytochrome c552 [EC 1.7.2.2]), first developed from anaerobic wastewater treatment reactors in 2004 (Mohan, et al. 2004), could be an effective approach to be used in such investigations. By checking the distribution and diversity of *nrfA* gene encoding DNRA in the sediments, it was found that
- 30 most of the *nrfA* sequences from the estuarine sediments of Clone River (UK) have several distinct clusters, including known nitrate ammonifiers, e.g., *Aeromonas, Shewanella, Desulfovibrio* and *Sulfurospillum* (Takeuchi, 2006). This is also the first trial to describe the *nrfA* partial sequences derived from a natural environment, with reference to their habitat-specific

community structure (Mohan, et al. 2004). However, most of the studies mentioned above focused only on the *nrfA* gene encoding DNRA, they paid less attention on the observable DNRA rate so far.

Some investigations have been also conducted for the sediment from different coastal areas in China. For example, the study showed that the N-loss as  $N_2$  was the main pathway in East China Sea (ECS), in where DNRA counts for 20–31% of benthic

- 5 nitrate reduction (Song et al., 2013). The investigation on the sediments from Yellow River Estuary (YRE) indicated that variability in the activities and community structure of DNRA bacteria (Cuina et al., 2017). There were several reports for showing the coexistence of denitrification and DNRA in Yellow River and Changjiang River in ECS. However, there are significant differences in both geographic location and climatic conditions between PRE and ECS or YRE. The PRE and ECS, as two largest estuaries in China in terms of freshwater discharge, are located in subtropical and temperate region.
- 10 respectively (Liu et al., 2018). YRE consists of typical turbid coastal waters that are influenced by the Yellow River which is the second largest river in the world in terms of sediment loadings, is located in monsoon and temperate region (Cui et al., 2011). Hong et al.'s work (Hong et al., 2019) indicated that estuaries play an important ecological role in the removal of overloaded nitrogen, thus relieving eutrophic pressure in coastal areas in PRE. To date there are lack of studies on the microbial mediated nitrate reduction pathways, especially DNRA that causes the active nitrogen delay in the PRE ecosystem.
- 15 Therefore, it is necessary to conduct a genetic investigation based on the sediments collected from PRE areas, in order to have a better understanding of DNRA pathway in PRE.

In this work, we chose HSE, one of the PRE, as the model area in the present investigation. The major focuses were to (1) examine the potential rates of DNRA in the sediments collected from the HSE continental shelf; (2) to develop the *nrf*A specific primer pairs for describing the distribution and phylogenetic status of the nitrate ammonifiers inhabited in the

20 sediments; and (3) to figure out the impact of the salinity and overloaded nitrogen on the DNRA. From which not only the environmental-gradient related DNRA profile would be revealed but also the activity and community compositions for the DNRA bacteria in these sediments were identified.

**2 Materials and methods**

**2.1 Details of the area selected**

- HSE (see Fig. 1) is referred to as Huangmao Bay with an area of approximately 540 km2. It is composed of a bay proper in the lower portion, a tidal river in the upstream and several island chains (Wei and Wu, 2011). The average water depth in the Bay is approximately 4.5 m. The Bay width changes from 35 km at the mouth to 1.9 km at the head, forming a funnel-shaped estuary (Wei and Wu, 2001). The Bay mouth features a chain of rock islands (Gaolan, Hebao and Dajing Islands) that provides a shelter from wave attack, resulting in an ideal place for port construction and tourism development. There
- 30 exists a deep navigation channel extending from the Bay mouth towards the Bay head, with water depths ranging from 6 to 22 m. The studied sites were located in the Bay mouth, around Dajing Islands. Sites C16-20 were near the west boundary of

the PRE Chinese White Dolphin National Nature Reserve. C3 was in the scope of influence of warm drainage pige and domestic sewage discharge from an factory.

**2.2 Sample collection and preparation**

Nine (9) sediments samples located in the sites C1, C3, C5, C8, C11, C13, C15, C16, and C17 were collected at twenty sites

- 5 (Fig. 1) during a cruise from Jun 8 to 22, 2016. These sediments samples were used for measuring the potential DNRA rate. Each sampling series involved taking three sediment cores from every station. A detailed description of these sites were listed in Table 1. Undisturbed sediment cores were collected by means of plexiglass tubes, driven directly into the sediment. After closing the tubes with a rubber stopper at the bottom, they were placed in a thermostatic container (which was enabled to preserve natural conditions) and transported from the sampling sites to the laboratory. During transportation, the container
- 10 was filled with a 20-centimeter layer of overlaying water from the sediment, as the source of oxygen and nutrients (An, et al. 2002).

At laboratory, each intact core was installed into a continuous-flow system (Fig. 2) consisting of an intake water vessel, flow tubes, intact sediment core, peristaltic pump, incubation bath, and sample collection vessel (Wayne, et al., 1995). A plunger, with Teflon inlet and outlet tubes and sealed to the core cylinder with an o-ring, was installed over each sediment core and

15 adjusted to 5 cm above the sediment surface to leave an overlying water volume (~ 230 mL). Each core was incubated at near in situ temperature and under laboratory light conditions, and site bottom water was passed continuously over the core surface with a flowrate of 1.5 mL min-1. After 24-h, the systems was allowed to approach steady-state conditions, and the inflow and outflow samples were collected daily for analysing the contents of dissolved gas and nutrient. After spiked 50 mg of K15NO3- into theflow water, the inflow and outflow water were sampled for the gas and nutrient analyses at 12, 24, 48, 72,

20 96, 120, 144, 168 and 216 h, respectively.

**2.3 Chemical analysis**

Concentrations of  ${}^{28}N_2$ ,  ${}^{29}N_2$ ,  ${}^{30}N_2$  and Ar were measured with GasBench-isotope ratio mass spectrometry (GasBench-IRMS, CTCAnalyticsAG, Zwingen, Switzerland), which had analytical capabilities of both dual-inlet and continuous-flow modes coupled with an Agilent 6890 GC equipped with an autosampler (CTC Analytics CombiPAL), to differentiate denitrification of internal sources of N versus that derived from the added  ${}^{15}NO_3^-$  in the overflowing water. The

25 denitrification of internal sources of N versus that derived from the added  ${}^{15}NO_3^-$  in the overflowing water. The concentrations of  ${}^{29}N_2$  and  ${}^{30}N_2$  were calculated according to Holtappels et al.' work (Holtappels, et al., 2011). Total NH4+ and  ${}^{15}NH_4^+$  (potential DNRA) production rates were measured by HPLC (Wayne, et al., 1995).

Environmental parameters including temperature, pH and salinity were on-site measured by a pH metre (PhS-3C, Rex Electric Chemical, China) and a salinometer (5150, San-xin Instrument, China). The concentrations of  $NO_3^-$ , DIN,  $NH_4^+$ ,

30 active P in water and Corg, P, N in the sediment were determined using an ultraviolet-visible spectrophotometer (8453 UV-VIS, Agilent, USA).

**2.4 Rate calculations**

The denitrification rates of the added 15NO3- (D15) and *in situ* 14NO3- (D14) were calculated from the measured production of 14N15N ( $p^{29}N_2$ ) and 15N15N ( $p^{30}N_2$ ): D15= $p^{29}N_2 + 2p^{30}N_2$ ; D14 = ( $p^{29}N_2/2 p^{30}N_2$ )·D15 (Gruca-Rokosz, et al.,2009). The total *in situ* rate of dissimilatory 14NO3- reduction to NH4+ (DNRA) was estimated from the production rate of

- 15NH4+ (p15NH4+) in an anoxic–anaerobic NO3- reduction zone. Assuming that DNRA takes place in the same sediment stratum as denitrification, the 15N atom% of NO3- reduced to NH4+ is the same as the 15N atom% of NO3- reduced to N2. The in situ rate of DNRA can therefore be calculated as: DNRA = p15NH4+ (D14/D15) (Gruca-Rokosz, et al.,2009, Giblin, et al., 2010). Further details are given by Risgaard–Petersen and Rysgaard (Risgaard-Petersen, and Rysgaard, 1995). The partitioning of DNRA and N2 production was expressed as %DNRA, i.e., = 100[DNRA/(DNRA + N2 production)], with total
- 10  $N_2$  production in the model or  $p^{30}N_2$  and  ${}^{29}N_2$  in the vials accounting for the remainder (Hardison, et al. 2015). The significance of the rate differences in the continuous-flow system were evaluated by the t-test or the Mann–Whitney Rank Sum Test, using a statistical software package (SigmaStat 3.5).

**2.5 Molecular biology analysis and DNRA communities**

DNA Extraction from the sediments. Bulk sediment samples were collected in the sites C1, C3, C5, C8, C11, C13, C15,

- 15 C16 and C17 from upper (0 3 cm) layers using an ethanol washed spatula, and stored in the box filled with crashed ice. DNA was extracted from wet sediment samples (1.0 g) using the PowerSoil DNA Kit (MO BIO Laboratories, USA) according to manufacturer's instructions, and the concentrations were measured on a spectrophotometer (E-Gel Imager, U.S.).
- Design of PCR primers. Primer pairs (6F/5R and 6F/4R) were designed based on the existing partial *nrf*A sequences and
   complete sequences including the nrfA fragment in an open reading frame (ORF). New pairs of forward and reverse primers (6F/5R) were manually designed based on the guidelines (Alkami Biosystems 1999) and the ClustalW (Thompson et al. 1994) multiple DNA sequence alignment. Two conserved regions of the nrfA sequences, consisting of one based on six sequences (AE008900, AL627282, AE005640, X72298, U32787, AE006035) and the other based on four sequences (AE005640, X72298, AE008900, AL627282) were chosen as the target of designing the new primer pairs, 6F/5R,
- 25 respectively. Another primer set (6F/4R) developed by Takeuchi (2016) was also employed after the PCR condition was optimized.

Sequences of the nrfA primers used in this study are listed in Table 2. Compared with Primer pairs (6F/4R), the PCR produces of primer pairs (6F/5R) had more conservative sequences for the high-throughput sequencing of nrfA gene in HSE Estuary samples.

30 **High-throughput sequence of** *nrf***A gene**. The DNRA community diversity of DNRA bacteria was determined on an Illumina MiSeq sequencer. Sites C1, C3, C5, C8, C11, C13, C15, C16 and C17 were selected to examine the community structure. After DNA extraction, PCR was conducted in triplicate using PCR Amplifier 2720 under the same conditions as

those reported previously (Song, et al. 2014). The PCR amplification was performed using a programmable thermocycler, Gene Amp 9700 (Applied Biosystems). The programmers consisted of the following conditions: the initial denaturation at 94°C for 2 min, 30 cycles of 58°C for the first-round PCR (20 cycles for the second-round PCR) for 1 min, followed by an annealing temperature at 46.5°C for the first-round PCR and at 54.5°C for the second-round PCR.

- 5 Sequence analysis and phylogenetic classification. PCR result was quantified using a PicoGreen dsDNA Assay Kit (Termo Scientifc, USA) on the TBS-380 Mini-Fluorometer (Turner Biosystems, CA, USA). Then, paired-end pyrosequencing was conducted using MiSeqReagent Kit V3 (Life Sciences, Branford, CT, USA) by the Illumina MiSeq sequence platform at the RayBiotech (Guangzhou, China). Raw reads of *nrf*A gene sequences were saved in the FsatQ file. First, the sequences shorter than 150 bp and lower than 20 quality score, as well as the sequences containing ambiguous base
- 10 N, were removed through the sliding window method. Paired reads were merged based on the overlap of bases using FLASH (http://ccb.jhu.edu/software/FLASH/) (Magoc., 2011). Then, the valid sequences were obtained through the identification between the merged reads and the relevant samples. Chimera sequences were removed using QIIME (http://qiime.org/) (Edgar, 2010) to get the high quality sequences, which were clustered into OTUs using UCLUST56 with 97% similarity and rare OTUs containing lower than 0.001% of total *nrfA* sequences were omitted (Edgar, 2011). The sequences with highest
- 15 abundance of each OTU as representative sequences were annotated with NCBI taxonomy using QIIME (soft version 1.9.1). Microbial community richness indices included Chao1 estimations, while diversity was quantified using the Simpson and Shannon indexes (Edgar, 2011). The alpha diversity index of each sample was determined using QIIME after a flattening process sequence (Page, 1996). Rank abundance curve was drawn using R software. The length of X-axis represented richness and slope of lines represented evenness. A steep gradient indicated low evenness. The differences of community
- 20 composition among four sites along the salinity gradient were evaluated through PCA based on community composition at genus level using QIIME software. Distributions of DNRA bacteria in four samples at different levels were determined using QIIME, and a heat map was constructed with the 50 most abundant genera using R software. In addition, RDA was conducted to evaluate the relationship between environmental factors and community structures using R software. Phylogenetic trees were generated using the neighbour-joining method as implemented with a TreeView programme (Page, 1996).

**3** Results and discussion**

**3.1 Environmental Gradients of the Estuary**

Bottom seawater and sediment characteristics were investigated at 20 stations (Fig. 1). Twenty (20) stations in the sites C1-C20 had been sampling quarterly for a one-year period during 2016. The national sea water quality standard in China (NO.

30 GB3097-1997), approved by the Ministry of Environmental Protection, was used in assessing the sea water quality of HSE. The Standards classified the seawater quality into four categories, and each grade of water is served for different purposes. The fourth grade of sea water are suitable for the marine development zones. The standard dissolved inorganic nitrogen (DIN) is referred to the totality of  $NO_3^-$ ,  $NO_2^-$  and  $NH_4^+$ . It showed that 80% data of  $NO_3^-$ , DIN and  $NH_4^+$  in the sites C1-C17 exceeded the four grade. While the areas in the sites C18-20, the water had better quality, which reached at the third grade. The sites C18-20 are located in near the PRE Chinese White Dolphin National Nature Reserve, and land pollutants were not so easy to spread in the area. As showed in Fig. 3a, DIN decreased sharply from the estuary coast (~ 0.842 mg/L) to the

- 5 outer shelf (~0.270 mg/L). In the same time, NO3- range in 0.17 0.567mg/L, which accounted for 60 80% of DIN. The concentrations of DIN in about 80% sites excessed 0.5 mg/L. The maximum concentration of DIN appeared at the mouth of Yamen Gate, decreases from upstream and estuarine coast to downstream. In general, the concentration of DIN is higher in the west and lower in the east, as shown in the profile figure. The highest concentration in the upper estuary area was above 0.95 mg/L, while the NID near the western estuary was above 0.8 mg/L and it declined from estuary to outer shelf. Fig. 3b
- 10 showed that the major form of DIN is  $NO_3^-$ , indicating that the pollutants in the western estuary was more serious than that of eastern area due to the hydrodynamic force. Figure 3c shows the rising trend in the concentration of  $NH_4^+$ , except for those observed in the sites C19-20. According to the monitoring from one-year period, the concentrations of  $NO_3^-$ , DIN and  $NH_4^+$  were lower during the spring than summer and winter. As was shown in Figure 4, the salinity was higher in the east than the west, and it is inclined to the west, especially in flood season. The result of salinity characteristics was consistent of
- 15 the hydrodynamic characteristics of the eastern tidal current in HSE. In flood season, the salinity was affected by fresh water runoff, so it was less than in the dry period. The salinity in our studied area was near the salinity contour (15000-17000 cl- mg/L) in bottom water.

**3.2 The potential rates of DNRA**

15NH+4 was detected in all sites after 12h in continuous-flow systems, which showed that DNRA existed in continuous-flow system. 15NH+4 ranged from 310.0 to 1948.5 ug/L after 216 h's experiments. The in situ potential rate of DNRA was different in every sampling sites. Moreover, the maximum 15NH+4 was 1948.5 ug/L in the site C3, the corresponding in situ rate of DNRA was 23.3 ug/(L.h) in 25h, and 6.27ug/(L.h) in the end of 216 h. The second maximum 15NH+4 was 1703.1 ug/L in the site C17, the corresponding in situ rate of DNRA was 2.75 ug/(L.h). While the low 15NH+4 was 310.0 ug/L, 398.2 ug/L in the sites C8 and C16, accordingly in situ rate of DNRA was 1.53 and 2.36 ug/(L.h). The in situ rate of DNRA 25 increases with the growth of cultivation time and reaches at the stable in five days, in which peak value was about 23.3

- ug/(L.h), in one to three days. After five days, the rate was ten slowed down to the range of 0.28 13.4 ug/(L.h).
  Sample sites have different hydrodynamic and pollutant characteristics, the sites C1 and C11 were in the upstream runoff channel, exposing to upstream pollutants, while the sites C5 and C7 were relatively near the coastline and the site C3 was the nearest these domestic sewage pipes and warm drainage (below sea level). As were shown in Figs. 5a and 5b, the potential
- 30 DNRA rates in the spatial distribution prove the insignificant statistical differences between the particular stations in salinity range, indicating the high potential DNRA rate near the sewage discharge in western coastal. It is believed that the most important factors effected DNRA were related to the real environment of the functional microorganisms. Furthermore, the

temperature, salinity, nitrogen and phosphorus may also be the important factors regulating the fate of  $NO_3^-$ . Due to the role of DNRA in nitrogen conversion, it could be included in the active nitrogen pollution pathway in the southern China.

**3.3 DNRA rates and nrfA gene abundance**

The abundances of DNRA bacteria based on *nrfA* gene quantification in the sediments of 9 sampling sites in Supplementary

- 5 Table S1. The two *nrfA* primer (6F/4R) and (6F/5R) were used in high-throughput sequencing respectively. Effective tags of 6F/5R *nrfA* primer are in the range of  $11 \times 10^6$  to  $7 \times 10^5$ , while that of 6F/4R *nrfA* primer is in range of  $7 \times 10^5$  to 9. Therefore, the analysis of DNRA community should be based on 6F/5R *nrfA* primer. High contents of *nrfA* gene fragments were found at the site C3, with values of 1159715 copics/(g wet sediment). Correspondingly, very low contents of *nrfA* were found at the sites C8, C16 and C15.
- 10 The potential DNRA rates was ranged from 0.13 ug/L.h to 23.3 ug/L.h in 72 hours. The highest DNRA rate was found in the site C3, which is located near coastal pollution zone. The lower rates were found in the sediment sampled at the sites C16 and C18, which is located near the White Dolphin National Nature Reserve. The sites C16 and C18 have lower DNRA rates, less than 3 ug/L.h in 72 hours. The maximum 15NH+4 was 1948.5ug/L in the site C3, the corresponding in situ rate of DNRA was highest in nine sites. Thus, potential DNRA rate and *nrfA* gene fragments had a positive correlation.

**15 **3.4 Spatial Distribution Patterns in the Sediment**

30

Via high-throughput sequencing, raw sequences were obtained from the sediment samples of PRE. After the removal of low quality reads, a total of 80,407–1,196,456 qualified sequences were produced. Then, a total of 74,173–1,159,715 effective sequences for these samples were obtained via further filtering, which were clustered into operational taxonomic units (OTUs) ranging from 2,541 to 8,709 (Supplementary Table S2). As shown in this table, the site C3 shared the highest

- 20 number of OTUs than any other sites, while the sites C8 and C16 shared few OTUs with other sites. Alpha diversity indices including Rarefaction Curves, Chao1 Curves, Shannon Curves, Simpson Curves and rank abundance Curves are listed in Supplementary Fig. S1. The Chao 1 estimators showed that the highest total species richness occurred at the site C3, while the lowest richness occurred in the farthest from the coast at the site C16. Both the Simpson index and the Shannon index of the sediment sampled at the sites C15 and C5, C17, C13 were higher than those of the other sites, indicating that the higher
- 25 community diversity was found in these four sites, although the difference was not substantial. Similarly, the same trend of sequence evenness with community richness was found (Fig. S1-e). The line representing the site C3 was the smoothest with the highest total species richness, and the site C15 had the highest community evenness. To compare the DNRA communities of nine sites, the principal components analysis (PCA) based on community

composition at genus level was performed (Hamady, 2010). Figure 6a shows the spatial variation of nine samples with 54.36% (PC1) and 18.15% (PC2) of the variance explained. Figs. 6a and 6b show that the sites C5 and C15 were clustered together,

and the community structures of these two sites were similar. The community composition of the site C11 also shared some

similarities with those of the site C17. The sites C3, C1, C8 and C16 were divided in one group in some similarities. However, the community of the site C13 was the most distinct among all the samples. The distribution pattern of the *nrf*A gene sequences from the surface sediments (0 - 1 cm) in HSE estuary was investigated. As shown in Figure 6b, the nrfA sequences distributed in the nine sites covering a salinity range of 10000 - 20000 cl- mg/L,

- 5 formed three distinct clusters corresponding to their geographical locations. Figure 6 shows that the nine samples in PCA divided in three groups were collected in the same sampling place. The sequences collected from the sites C1, C11 and C13 were clearly separated from the other cluster. The sites C8 and C16 were just in the same salinity contours, which were still tilted west. This result suggests that the salinity per se was likely to affect the selection of the *nrfA* bacterial populations. Since the *nrfA* sequences had been so far extracted from various sources (bioreactors and estuarine sediments) as well as
- 10 from laboratory cultures beyond the phylogenetic groups ( $\alpha$ -, $\gamma$ -,  $\delta$ -,  $\varepsilon$ -proteobacteria and the Bacteroides-related subdivisions), it was quite reasonable to detect the similar *nrfA* sequences of different phylogenetic origins in the environmental samples (e.g., one from the *Enterobacteriaceae* and the other from the sulphate reducing bacteria). This approach makes it possible to detect the functional marker in situ but may reduce its analytical resolution into the community structure. As a result, the *nrfA*-based grouping in the present study is not able to distinguish the freshwater from seawater
- 15 sequences. However, it was successful in extracting the unique estuarine sequences.

**3.5 Diversity of the nrfA Sequences in the Sediments**

After removing rare OTUs, modified OTUs were classified into different levels such as phylum, class, order and so on. The classification results are listed in Supplementary Table S3. In this study, a total of 7 bacterial phyla were identified. Dominant phyla (> 0.5% sequence abundance in at least one site) are listed in Fig. 7a. For bacterial phyla, *Proteobacteria*

20 was the most abundant phylum in all samples, accounting for 99.93 - 91.56%. In addition, many sequences based on *nrfA* pyrosequencing were not classified to certain type, range from 8.1% in the site C5 to 0.07% in the site C16. They appeared in sediment samples from home and abroad.

At the class level, as shown in Fig. 7b, a total of 8 classes in eight sites were obtained. *Gammaproteo* bacteria was the most abundant of all sites, accounting for 99.9 – 90.5%. Other *Proteobacteria* such as *Alphaproteo* bacteria and *Deltaproteo*

- 25 bacteria are existed, but less than 0.5%. As depicted in Fig. 7c, *Enterobacteriales* (89.9 99.9%) which belong to *Gammaproteo* bacteria was the most abundant order. Of all filtered sequences, a total of 11 families were detected. *Enterobacteriaceae*, which is member of *Enterobacteriales*, were more abundant than other families, accounting for 89.9 99.0% of the classified sequences. *Microbulbiferaceae* belonging to *Cellvibrionales, Moraxellaceae* belonging to *Pseudomonadales*, and other 7 families were present in the site C5 at low levels. While other sites have only 1-3 families.
- 30 An overview of the diversity of the *nrfA* sequences in the HSE sediments was also examined. A total of 63 sequences retrieved from the sequences found in the sediments were compare with those from known strains and environmental isolates. Fig. 8 showed a dendrogram, suggesting the relationship of partial nucleotide sequences form the *nrfA* sequences and the other reference sequences obtained from the databases.

Three nucleotide sequence clusters were roughly grouped: One cluster belonged to the known group as nitrate ammonifiers with the *nrfA* sequences, comprised of fermentative bacteria (*Escherichia coli*) (Cole, et al., 1980) and the others. Most of the *nrfA* sequences from the HSE sediments were closely related to the *nrfA* sequences from *Shewanella*, and *Sulfurospirillum*, which are known members of the nitrate ammonifiers (Takeuchi, 2006). A few of the sequences were quite

- 5 divergent from the others and affiliated with the *nrfA* sequences retrieved from versatile anaerobes, such as sulphur-reducing nitrate ammonifier *Sulfurospirillum deleyianum* (Eisenmann, et al. 1995). These microbes seem to be favoured to anaerobic marine sediments full of iron sulphide, involved in sulphur and iron cycles coupled with nitrate ammonification. Although the present result partly supports the previous view on the contribution of fermentative bacteria to nitrate ammonification, it also suggested the majority of the actual agents could be the other metabolically versatile but less described groups, such as
- 10 *Shewanella*, *Sulfurospirillum* and *Bacteroides* (Cuina, et al., 2017, Takeuchi, 2006). Widespread distribution of these *nrfA* sequences found in this study could be a result of their adaptation to nitrate and organics rich habitats. The nitrate ammonifying populations in the sediment have been so far treated as a minor component but ecologically important not only as a consumer of nitrite but also as a producer of ammonium, while some formed unique clusters grouped with *Dusulfovibrio salexigens* or supposedly uncultured members of bacteria.

**15 4 Conclusion**

This study investigated the existence of DNRA using the 15N isotope paring method in continuous-flow systems, based on the 20 sites of HSE during a one-year period. The results showed that the maximum  $^{15}NH_4^+$  was 1948.5 ug/L, the corresponding in situ rate of DNRA was 6.3ug/L.h in 72h. High contents of *nrfA* gene fragments were found at the same site, with values of 1159715 copics/(g wet sediment). Correspondingly, very low contents of *nrfA* were found at other sites. The

- 20 distribution pattern of the *nrfA* gene sequences from the surface sediments formed three distinct clusters corresponding to their geographical locations. which suggests that salinity per se was likely to affect the selection of the *nrfA* bacterial populations. Base on the nrfA gene pyrosequencing, *Enterobacteriaceae* (a member of *Enterobacteriales*, were more abundant than other families) counts for 89.9 99.0% of the classified sequences. This study improves our understanding of DNRA in the PRE. However, the further study is required to understand the contributions of anammox, denitrification and
- 25 DNRA to nitrate reduction.

**Acknowledgments**

This study is funded by the National Natural Science Foundation of China (Grant no. 51409287) and the Guangdong Province Natural Science Foundation of China (Grant no. 2017A030313329).

**References**

Pan, J.Y., Gu,Y.Z., Wang,D.X.: Observations and numerical modeling of the Pearl River plume in summer season, J. Geophys. Res. Oceans, 119, 2480-2500, https://doi.org/10.1002/2013JC009042, 2014.

Dai,M., Wang, L., Guo, X., Zhai, W., Li, He, Q. B. and Kao, S.J.: Nitrification and inorganic nitrogen distribution in a
large perturbed river/estuarine system: the Pearl River Estuary, China, Biogeosciences, 5, 1227-1244, https://doi.org/10.5194/bg-5-1227-2008, 2008

Hong, Y.G., Wu, J.P., Guan, F. J., Yue, W.Z., Long, A.M.: Nitrogen removal in the sediments of the Pearl River Estuary, China: Evidence from the distribution and forms of nitrogen in the sediment cores, Mar. Pollut. Bull., 138, 115-124, https://doi.org/ 10.1016/j.marpolbul.2018.11.040,2019.

- Wei, X. and Wu, X. X.: Dynamic structures and their sedimentation effects of Yamen Inlet, Huangmaohai estuary, China Earth Sci., 54, 937-945, https://doi.org/10.1007/s11430-010-4106-9, 2011.
   Liu, J.N., Du, J.Z., Wu, Y., Liu, S.M., Nutrient input through submarine groundwater discharge in two major Chinese estuaries: the Pearl River Estuary and the Changjiang River Estuary, Estuarine, Coastal Shelf Sci., 203,17-28, https://doi.org/10.1016/j.ecss.2018.02.005, 2018
- Cui, B.L.,Li, X.y.,Coastline change of the Yellow River estuary and its response to the sediment and runoff (1976–2005),Geomorphology, 127,32-40, https://doi.org/10.1016/j.geomorph.2010.12.001,2011
   Bulletin of Marine Quality in Jiangmen City, http://hyyyj.jiangmen.gov.cn/zwgk/gb/201809/t20180928\_1683848.html, last access:12 October 2018
   U. h. D. ta Miran and Kanaka and Kanaka

Herbert, R. A.: Nitrogen cycling in coastal marine ecosystems marine ecosystems. FEMS Microbiol Rev..23, 563-90, 1999

- 20 Megonigal, J. P., Hines, M. E. and Visscher, P. T., Anaerobic metabolism: linkages to trace gases and aerobic processes. In: Biogeochemistry, edited by: Schlesinger, W.H., Elsevier-Pergamon, Oxford, 317–424, 2004. Tjedje, J. M., Ecology of denitrification and of dissimilatory nitrate reduction to ammonium. In: Biology of anaerobic microorganisms, edited by: Zehnder, A. J. B., John Wiley and sons, Inc., New York, NY, 179-244, 1988. Laverman, A. M., Canavan, R. W., Slomp, C. P. and Cappellen, P. V. : Potential nitrate removal in a coastal freshwater
- 25 sediment (Haringvliet Lake, The Netherlands) and response to salinization, Water Res. 41, 3061-3068, https://doi.org/ 10.1016/j.watres.2007.04.002, 2007.

Hardison, A. K., Algar, C.K., Giblin, A. E., and Rich, J. J., Influence of organic carbon and nitrate loading on partitioning between dissimilatory nitrate reduction to ammonium (DNRA) and N2 production, Geochim. Cosmochim. Ac., 164, 146-160, http://dx.doi.org/10.1016/j.gca.2015.04.049, 2015.

30 Gardner, W. S. and McCarthy, M. J., Nitrogen dynamics at the sediment–water interface in shallow, sub-tropical Florida Bay: why denitrification efficiency may decrease with increased eutrophication, Biogeochemistry, 95: 185–198. http://dx.doi.org/10.1007/s10533-009-9329-5, 2009. Song, G.D., Liu, S.M., Marchant, H., Kuyper, M. M. M. and Lavik, G., Anammox, denitrification and dissimilatory nitrate reduction to ammonium in the East China Sea sediment, Biogeosciences, 10: 6851-6864, http://dx.doi.org/10.5194/bg-10-6851-2013, 2013

Cuina, B., Wang, Y., Ge, C., Ahmad, H. A., Gao, B. and Ni, S. Q., Dissimilatory Nitrate Reduction to Ammonium in the

5 Yellow River Estuary: Rates, Abundance, and Community Diversity, Sci. Rep., 7:1-11, https://doi.org/10.1038/s41598-017-06404-8, 2017
Control of the Matter of the Ma

Gardner, W. S., McCarthy, M. J., An, S. and Sobolev, D., Nitrogen fixation and dissimilatory nitrate reduction to ammonium (DNRA) support nitrogen dynamics in Texas estuaries. Limnol. Oceanogr., 51:558–568, https://doi.org/10.4319/lo.2006.51.1\_part\_2.0558, 2006

- Smith, C. J., Nedwell, D. B., Dong, L.F. and Osborn, A. M., Diversity and abundance of nitrate reductase genes(narG and napA), nitrite reductase gnes(nirS and nrfA), and their transcripts in estuarine sediments. Appl. Environ, Microbiol., 73: 3612-3622, https://doi.org/10.1128/AEM.02894-06, 2007 Takeuchi, J., Habitat segregation of a functional gene encoding nitrate ammonification in estuarine sediments, Geomicrobiology Journal, 23: 75-87, http://dx.doi.org/10.1080/01490450500533866, 2006
- 15 Mohan, S.B., Schmid, M., et al. Detection and widespread distribution of the nrfA gene encoding nitrite reduction to ammonia, a short circuit in the biological nitrogen cycle that competes with denitrification. FEMS Microbiol. Ecol. 49: 433-443 https://doi.org/10.1016/j.femsec.2004.04.012, 2004, An Soonmo, Gardner Wayne S. Dissimilatory nitrate reduction to ammonium (DNRA) as a nitrogen link, versus

denitriffication as a sink in a shallow estuary( laguna Madre/Baffin Bay, Texa), Marine Ecology Progress Series, 237:41-52

IF2.3 doi:10.3354/meps237041,2002.
Holtappels, M., Lavik, G., Jensen, M. M., & Kuypers, M. M. M.: 15N-labeling experiments to dissect the contributions of heterotrophic denitrification and anammox to nitrogen removal in the OMZ waters of the ocean, Methods in Enzymology. 486, 223–251, https://doi.org/10.1016/B978-0-12-381294-0.00010-9 ,2011.
Wayne, S. G., Bootsma, H.A., Evans, C., John, P.A.S., Improved chromatographic analysis of 15N:14N ratios in ammonium

or nitrate for isotope addition experiments, Marine Chemistry 48 271-282 DOI: 10.1016/0304-4203(94)00060-Q(1995)
 Gruca-Rokosz, R., Tomaszek, J. A., Koszelnk, P. Competitiveness Of Dissimilatory Nitrate reduction Processes In Bottom Sediment Of Rzeszów Reservoir, Environment Protection Engineering, 35 (2) ,5-13, 2009, IF0.486
 Giblin, A.E., Weston, N.B., Banta, G.T. The effects of salinity on nitrogen losses from an Oligohaline Estuarine sediment, Estuaries and Coasts, 33:1054-1068 IF 2.42 DOI 10.1007/s12237-010-9280-7, 2010

30 Risgaard-Petersen, N., Rysgaard, S., Nitrate reduction in sediments and waterlogged soil measured by 15N techniques, Methods in applied soil microbiology and biochemistry, Edited by Alef K., Nannipieri P., Academic Press, London, 1995, 279–288 Song, B., Lisa, J. A. & Tobias, C. R. Linking DNRA community structure and activity in a shallow lagoonal estuarine system. Frontiers in Microbiology. 5, 460, 2014. IF 4 doi: 10.3389/fmicb.2014.00460 Magoc., T. and Salzberg., S. FLASH: Fast length adjustment of short reads to improve genome assemblies. Bioinformatics 27:21 2957-63. doi: 10.1093/bioinformatics/btr507, 2011.

Caporaso, J. G. Kucaynski, J. Stombaugh, J. OIIME allows analysis of high-throughput community sequencing data. Nat 5 Met 7, 335–336. DOI:10.1038/nmeth.f.303 (2010) Edgar, R. C. Search and clustering orders of magnitude faster than BLAST. Bioinformatics 26, 2460–2461. doi:10.1093/bioinformatics/btg461 (2010) Edgar, RC, Haas, BJ, Clemente, JC, Quince, C, Knight, R. 2011. UCHIME improves sensitivity and speed of chimera detection, Bioinformatics. 27(16), 2000-2194 DOI: 10.1371/journal.pone.0098115

Page, R. D.M. 1996. TREEVIEW: an application to display phylogenetic trees on personal computers. Comp Appl Biosci 12:357-358

Hamady, M., Lozupone, C. & Knight, R. Fast UniFrac: facilitating high-throughput phylogenetic analyses of microbial communities including analysis of pyrosequencing and PhyloChip data. ISME J. 4, 17-27 (2010).

https://doi.org/10.1038/ismej.2009.97 15

10

20

Cole JA, Brown CM. 1980. Nitrite reduction to ammonia by fermentative bacteria: a short circuit in the biological nitrogen cycle. FEMS Microbiol Lett 7:65-72. DOI: 10.1111/j.1574-6941.1980.tb01578.x

Eisenmann E, Beuerle J, Sulger K, Kroneck PMH, Schmacher W. 1995. Lithotrophic growth of Sulfurspirillum deleyianum with sulphide as electron donor coupled to respiratory reduction of nitrate to ammonia. Arch Microbiol 164:180–185. DOI: 10.1007/BF02529969

| Site ID    | Salinity
psu | рН   | $NH_4^+$               | Active
P ug/L | $NO_3^-$ | Corg
%        | N
mg/kg | Р   |
|------------|-----------------|------|------------------------|------------------|----------|------------------|------------|-----|
|            |                 |      | Surface water (0-50cm) |                  |          | Sediment (0-5cm) |            |     |
| C1         | 0.3             | 7.76 | 1.3                    | 42               | 633      | 1.70             | 1017       | 867 |
| C3         | 1.0             | 7.96 | 1.9                    | 32               | 622      | 1.25             | 896        | 835 |
| C5         | 12.9            | 7.98 | 2.2                    | 28               | 532      | 0.71             | 516        | 468 |
| C 8 | 13.4            | 8.12 | 3.3                    | 29               | 504      | 1.00             | 596        | 729 |
| C11        | 2.15            | 7.74 | 0.9                    | 32               | 495      | 1.37             | 897        | 732 |
| C13        | 9.55            | 7.61 | 0.9                    | 35               | 475      | 1.00             | 867        | 839 |
| C15        | 12.5            | 7.53 | 0.6                    | 34               | 494      | 1.38             | 861        | 523 |
| C16        | 9.9             | 7.66 | 1.1                    | 40               | 415      | 1.24             | 723        | 589 |
| C17        | 13.0            | 7.72 | 1.9                    | 29               | 418      | 0.98             | 689        | 368 |

**Table1. Water and sediment data in HSE**

Temperture in the area: 22-26 °C

**Table 2. Sequences of the nrfA primers used in this study 25**

| Primer                                | Sequence(5 ' to 3')     |  |  |  |
|---------------------------------------|-------------------------|--|--|--|
| 5R                                    | CGCCAYTGVGCRTGRCGRATATC |  |  |  |
| 6F                                    | GAYTGCCAYATGCCRAAAGT    |  |  |  |
| 4R                                    | GCATCCGCSGCTTTATCCAT    |  |  |  |
| F: forward primer, R: reverse primer; |                         |  |  |  |

Base Codes: K=T/G, R=A/G, Y=C/T, S=C/G

5

elevation(m) -1.0 -2.0 -3.0 -4.0 -5.0 -6.0 -7.0 -8.0 -9.0 -11.0 -12.0 -13.0 -14.0 -14.0 -12.0 -13.0 -14.0 -14.0 -10.0 -1.0 -2.0 -6.0 -7.0 -9.0 -1.0 -1.0 -6.0 -7.0 -1.0 -1.0 -1.0 -6.0 -7.0 -1.0 -1.0 -1.0 -6.0 -7.0 -1.0 -1.0 -1.0 -1.0 -1.0 -1.0 -1.0 -1.0 -1.0 -1.0 -1.0 -1.0 -1.0 -1.0 -1.0 -1.0 -1.0 -1.0 -1.0 -1.0 -1.0 -1.0 -1.0 -1.0 -1.0 -1.0 -1.0 -1.0 -1.0 -1.0 -1.0 -1.0 -1.0 -1.0 -1.0 -1.0 -1.0 -1.0 -1.0 -1.0 -1.0 -1.0 -1.0 -1.0 -1.0 -1.0 -1.0 -1.0 -1.0 -1.0 -1.0 -1.0 -1.0 -1.0 -1.0 -1.0 -1.0 -1.0 -1.0 -1.0 -1.0 -1.0 -1.0 -1.0 -1.0 -1.0 -1.0 -1.0 -1.0 -1.0 -1.0 -1.0 -1.0 -1.0 -1.0 -1.0 -1.0 -1.0 -1.0 -1.0 -1.0 -1.0 -1.0 -1.0 -1.0 -1.0 -1.0 -1.0 -1.0 -1.0 -1.0 -1.0 -1.0 -1.0 -1.0 -1.0 -1.0 -1.0 -1.0 -1.0 -1.0 -1.0 -1.0 -1.0 -1.0 -1.0 -1.0 -1.0 -1.0 -1.0 -1.0 -1.0 -1.0 -1.0 -1.0 -1.0 -1.0 -1.0 -1.0 -1.0 -1.0 -1.0 -1.0 -1.0 -1.0 -1.0 -1.0 -1.0 -1.0 -1.0 -1.0 -1.0 -1.0 -1.0 -1.0 -1.0 -1.0 -1.0 -1.0 -1.0 -1.0 -1.0 -1.0 -1.0 -1.0 -1.0 -1.0 -1.0 -1.0 -1.0 -1.0 -1.0 -1.0 -1.0 -1.0 -1.0 -1.0 -1.0 -1.0 -1.0 -1.0 -1.0 -1.0 -1.0 -1.0 -1.0 -1.0 -1.0 -1.0 -1.0 -1.0 -1.0 -1.0 -1.0 -1.0 -1.0 -1.0 -1.0 -1.0 -1.0 -1.0 -1.0 -1.0 -1.0 -1.0 -1.0 -1.0 -1.0 -1.0 -1.0 -1.0 -1.0 -1.0 -1.0 -1.0 -1.0 -1.0 -1.0 -1.0 -1.0 -1.0 -1.0 -1.0 -1.0 -1.0 -1.0 -1.0 -1.0 -1.0 -1.0 -1.0 -1.0 -1.0 -1.0 -1.0 -1.0 -1.0 -1.0 -1.0 -1.0 -1.0 -1.0 -1.0 -1.0 -1.0 -1.0 -1.0 -1.0 -1.0 -1.0 -1.0 -1.0 -1.0 -1.0 -1.0 -1.0 -1.0 -1.0 -1.0 -1.0 -1.0 -1.0 -1.0 -1.0 -1.0 -1.0 -1.0 -1.0 -1.0 -1.0 -1.0 -1.0 -1.0 -1.0 -1.0 -1.0 -1.0 -1.0 -1.0 -1.0 -1.0 -1.0 -1.0 -1.0 -1.0 -1.0 -1.0 -1.0 -1.0 -1.0 -1.0 -1.0 -1.0 -1.0 -1.0 -1.0 -1.0 -1.0 -1.0 -1.0 -1.0 -1.0 -1.0 -1.0 -1.0 -1.0 -1.0 -1.0 -1.0 -1.0 -1.0 -1.0 -1.0 -1.0 -1.0 -1.0 -1.0 -1.0 -1.0

10

Figure 1. Location of sampling stations in HSE

5

Figure 2. Diagram of Continuous-flow System